# OptiFluence: Scalable and Principled Design of Privacy Canaries

## Abstract

Privacy auditing has emerged as a practical tool for empirically estimating training data leakage in machine learning models (in contrast to the provable but often overly pessimistic bounds provided by a differential privacy analysis). A common strategy is to use membership inference attacks to detect the presence in training data of specific canaries (points designed to maximize memorization). However, existing canary designs are largely heuristic, relying on mislabeled or out-of-distribution samples. We address this gap by formulating canary design as a bilevel optimization problem, where the model is trained in the inner loop and the canary is optimized in the outer loop to maximize its detectability. Building on this view, we develop OptiFluence, a scalable framework that combines two components: (i) influence-based pre-selection to identify promising canary seeds; and (ii) unrolled sample optimization with memory-efficient gradient techniques. Our approach achieves remarkable empirical performance on two standard privacy auditing datasets, MNIST and CIFAR-10: optimized canaries demonstrate up to 415× higher detectability than in-distribution baselines, reaching near-perfect detection rates of 99.5% TPR at 0.1%FPR. Critically, these canaries transfer effectively across different model architectures without retraining, enabling practical third-party privacy audits. This transferability allows regulators and auditors to assess model privacy without requiring access to proprietary training infrastructure or substantial computational resources.

## 1 Introduction

Machine learning models can inadvertently leak sensitive information about their training data, raising significant privacy concerns in real-world applications (Carlini et al., 2021). Because it is difficult to obtain a realistic assessment of such privacy leakage in a practical ML pipeline using only natural data, researchers often insert *canaries*—artificially designed points intended to maximize memorization (Carlini et al., 2019a)—into training sets and then measure whether membership inference attacks (MIAs) can detect them.

Existing approaches to designing canaries, however, are largely heuristic and ad hoc. They typically rely on mislabeled examples (Nasr et al., 2023), out-of-distribution points (Meeus et al., 2025), or adversarial perturbations Wen et al. (2022). While such strategies are simple to deploy, they risk underestimating privacy leakage: if the chosen canaries are not maximally detectable, the resulting audit may provide a false sense of security.

In this work, we present a *systematic* and *principled* approach to optimizing privacy canaries. We formulate canary construction as a *bilevel optimization problem* that explicitly maximizes the likelihood ratio for membership inference—moving beyond heuristic designs toward a rigorous mathematical foundation. This formulation allows us to construct canaries that are provably more detectable than existing approaches.

We instantiate this framework through OptiFluence, which combines two complementary strategies: *influence-based pre-selection* to identify promising initial canaries from natural data, and *gradient-based sample optimization* using unrolled training dynamics to fine-tune these candidates. To make this approach scalable, we develop memory-efficient techniques that take advantage of *rematerialization* (gradient checkpointing) and truncated backpropagation through time Williams & Peng (1990).

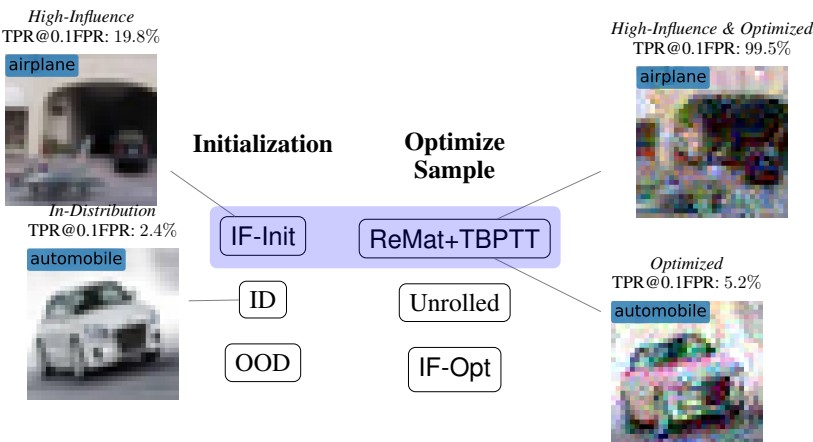

Figure 1: **Canary Optimization Pipeline** involves two stages of initialization and sample optimization. OptiFluence (highlighted) picks the strongest primitives at each stage: a robust initialization based on influence (IF-Init) and a performant optimization strategy (ReMat+TBPTT)

Our optimized canaries achieve remarkable performance improvements: up to $415\times$ higher detectability than in-distribution baselines and near-perfect detection rates (99.5% TPR at 0.1% FPR) on standard benchmarks (see Figure 1 for two examples). Critically, these canaries transfer effectively across different model architectures—a property that enables practical *third-party auditing* without requiring access to the model provider's infrastructure or placing excessive computational demands on auditors such as regulators with limited resources.

**Our contributions are:**

- We cast canary design as a bilevel optimization problem with a formal privacy loss objective, providing a principled foundation that unifies and significantly outperforms prior heuristic constructions.

- We develop OptiFluence, a scalable optimization pipeline that solves the canary optimization problem through influence-based pre-selection and memory-efficient unrolled sample optimization with rematerialization and truncated backpropagation.

- We demonstrate empirically on MNIST and CIFAR-10 that optimized canaries achieve up to $415\times$ higher detectability than heuristic baselines while maintaining strong transferability across model architectures, enabling practical third-party privacy audits.

- We show that transferability enables efficient regulatory auditing by allowing auditors to require model providers to include pre-optimized canaries in training without needing access to proprietary infrastructure or substantial computational resources.

## 2 RELATED WORK

Prior work has shown that certain training samples are inherently more vulnerable to membership inference. For instance, Kulynych et al. (2022) found minority subpopulations to be disproportionately exposed, while Carlini et al. (2021) observed similar vulnerability among outliers. In vision models, Jagielski et al. (2020) created poisoned points to stress-test DP-SGD, and in language models, Carlini et al. (2019a) and follow-ups demonstrated that models can memorize synthetic "secrets" inserted into training text. More recent studies extend this line to large language models (LLMs), using identifiers, rare sequences, or synthetic text to probe memorization (Yue et al., 2023; Meeus et al., 2025; Panda et al., 2024). Together, these works highlight the existence of privacy-sensitive samples but rely on task-specific heuristics.

Beyond inputs, privacy canaries have also been constructed in parameter space. For example, Maddock et al. (2023) proposed weight-space and gradient-based canaries in federated learning, later generalized to randomized updates (Pillutla et al., 2023). Other studies in the federated setting

examined unintended memorization of user-specific features (Thakkar et al., 2020). These approaches broaden the notion of canaries but still depend on domain knowledge and manual design.

Perhaps the closest works to ours are Nasr et al. (2023) and Boglioni et al. (2025). Nasr et al. (2023) construct input-space gradients by searching for a canary whose gradient is orthogonal to the average in-distribution model update. We discuss this principle and its connection to OptiFluence and our baselines in Appendix D. Boglioni et al. (2025) uses metagradients (Engstrom et al., 2025) to calculate privacy canaries over training trajectories. Boglioni et al. (2025) however is focused on single-run auditing Steinke et al. (2023) which means many canaries get injected into the training run to simulate the effect of adding a single sample in the more traditional MIA attacks. By contrast, we only optimize a single canary, do not assume any particular auditing setup a-priori, and assume that the canary will be added to the training set and sampled randomly by the training algorithm. From a canary optimization prespective, OptiFluence subsumes Boglioni et al. (2025) as a special case by choosing an in-distribution sample and using rematerialized gradients. Our ablations in Section 6.3 covers this special case as well.

Finally, empirical auditing methods have matured into rigorous membership-inference frameworks. Early work asked whether DP-SGD was more private in practice than its theoretical guarantees (Jagielski et al., 2020), while later studies quantified memorization through canary extraction (Carlini et al., 2019a) and formalized likelihood-ratio testing via LiRA (Carlini et al., 2022). Recent evaluations showed that many defenses appear effective only because they ignore the most vulnerable points (Aerni et al., 2024).

In summary, while prior work demonstrates the usefulness of canaries for auditing, their design remains heuristic and ad hoc. Our contribution is to cast canary construction as an optimization problem, unifying these disparate efforts into a principled framework that yields stronger membership inference attacks.

## 3 BACKGROUND

### 3.1 INFLUENCE FUNCTIONS

Influence functions, a classical tool from robust statistics, were scaled by Koh & Liang (2017) to explain large ML models by identifying the "most influential" training samples for a prediction.

Let $D_{\text{train}} = \{z_i = (x_i, y_i)\}$ and $\mathcal{L}(\theta; z_i)$ be the per-sample loss. The trained model is the the solution to the empirical risk minimization $\theta^* = \text{argmin}_{\theta \in \mathbb{R}^d} \frac{1}{N} \sum_{i=1}^{N} \mathcal{L}(\theta; z_i)$. Assuming $\theta^*$ exists and is unique, consider upweighting a sample $z_j$ by $\alpha$: $\theta^*(\alpha) = \text{argmin} \frac{1}{N} \sum_i \mathcal{L}(\theta; z_i) + \alpha \mathcal{L}(\theta; z_j)$. By the Implicit Function Theorem, the parameter influence of $z_j$ is

$$I_{\theta^*}(z_j) \triangleq \frac{d\theta^*}{d\alpha}\big|_{\alpha=0} = -H^{-1} \nabla_\theta \mathcal{L}(\theta^*; z_j), \tag{1}$$

with $H = \nabla_\theta^2 \ell(\theta^*; D_{\text{train}})$ as the converged model $\theta^*$ Hessian. For small $\alpha$, $\theta^*(\alpha) - \theta^* \approx \alpha I_{\theta^*}(z_j)$.

To interpret influence on a sample $z_i$, we set $f(\theta) = \mathcal{L}(\theta; z_i)$. Its first-order change from perturbing $z_j$ is $I(x_i; x_j) \triangleq -\nabla_\theta \mathcal{L}(\theta^*; z_i)^\top H^{-1} \nabla_\theta \mathcal{L}(\theta^*; z_j)$, so that $\alpha I(x_i; x_j)$ approximates the change in $z_i$'s loss if $z_j$'s training weight is increased by $\alpha$. We provide a more detailed background in Appendix A.

### 3.2 DIFFERENTIAL PRIVACY

We characterize privacy leakage via the canonical notion of differential privacy (Dwork et al., 2006).

**Definition 1** (Differential Privacy). *A randomized mechanism $\mathcal{M}$ is $(\varepsilon, \delta)$-DP if for all datasets $D, D' \in \mathcal{D}$ differing in one datapoint and for all events $\mathcal{O}$: $P[\mathcal{M}(D) \in \mathcal{O}] \leq e^\varepsilon P[\mathcal{M}(D') \in \mathcal{O}] + \delta$.*

In the above definition, $\delta \in (0, 1)$ can be thought of as a very small failure probability, and $\varepsilon > 0$ is the privacy loss; smaller $\epsilon$ and $\delta$ correspond to stronger privacy guarantees. We focus on the information leakage of the mechanism $\mathcal{M}$ of releasing the model gradients (and hence intermediate checkpoints) during training, as widely studied with stochastic gradient descent (SGD).

Since the two datasets $D, D'$ can only differ in a single sample (according to Definition 1), as privacy auditors we seek to find the sample that gives them the best odds of detecting a change in the output

of the mechanism. To concretize this, let us forgo the probability of failure and set $\delta = 0$. The privacy parameter $\varepsilon$ bounds the *odds* of detecting a change:

$$\log \left\{ \frac{P[\mathcal{M}(D \cup \{x\}) \in \mathcal{O}]}{P[\mathcal{M}(D) \in \mathcal{O}]} \right\} \leq \varepsilon, \tag{2}$$

where, without loss of generality, we took $x = D' \setminus D$ to be the sample difference between $D$ and $D'$. We evaluate auditor's success using *membership inference attacks* (MIAs)—the canonical technique for empirical estimation of privacy leakage of ML models (Shokri et al., 2017; Jagielski et al., 2020).

A common MIA involves the auditor training **shadow models** with full knowledge of what samples were in the training set (IN), and which were not (OUT). Armed with this knowledge, the auditor seeks to predict membership a sample $x$ in a target model's training set and predict IN vs. OUT. The success of the auditor establishes a lower bound on the left-hand side of Equation (2)—i.e., a *lower bound on the privacy leakage for sample* $x$. In this work, we employ an improved MIA known as LiRA (Carlini et al., 2022) (see Algorithm 2 in Appendix B.4) which formalizes MIA as a hypothesis test of membership. We take advantage of this interpretation in Section 4.

## 4   CANARY OPTIMIZATION AS LIKELIHOOD RATIO MAXIMIZATION

Let $\mathcal{T}(D)$ denote training on dataset $D$. For a target sample $(x, y)$ define $Q_{\text{in}}(x, y) = \{ h \leftarrow \mathcal{T}(D \cup \{(x, y)\}) \mid D \sim \mathcal{D} \}, Q_{\text{out}}(x, y) = \{ h \leftarrow \mathcal{T}(D) \mid D \sim \mathcal{D}, (x, y) \notin D \}$ the distributions over trained models with or without $(x, y)$.

Given a model $h$, the adversary's task is to test

$$H_0 : h \sim Q_{\text{out}}(x, y), \qquad H_1 : h \sim Q_{\text{in}}(x, y).$$

By the Neyman–Pearson lemma, the most powerful level-$\alpha$ test rejects $H_0$ when the likelihood–ratio

$$\Lambda(h; x, y) = \frac{p(h \mid Q_{\text{in}}(x, y))}{p(h \mid Q_{\text{out}}(x, y))} \tag{3}$$

exceeds a threshold. Here $p(h \mid Q_b(x, y))$ is the probability density of $h$ under the distribution $Q_b(x, y)$ for $b \in \{\text{in}, \text{out}\}$.

In practice, the distributions $Q_{\text{in}}, Q_{\text{out}}$ are intractable. We therefore work with the *log–likelihood ratio* and restrict attention to statistics computable from model predictions:

$$\ell_{\text{priv}}(x, y) = \log p(y \mid h_{D \cup \{(x,y)\}}, x) - \log p(y \mid h_D, x), \tag{4}$$

where $h_D$ and $h_{D \cup \{(x,y)\}}$ are models trained on $D$ and $D \cup \{(x, y)\}$, respectively. Maximizing (4) corresponds to constructing the most distinguishable canary $(x, y)$ for the likelihood–ratio test.

We can make this optimization more numerically stable by directly working with pre-softmax logits. Let $g(\theta; x) \in \mathbb{R}^{|\mathcal{Y}|}$ denote the pre-softmax logits. Carlini et al. (2022) showed that

$$\log p(y \mid \theta, x) = g(\theta; x)_y - \log \sum_{y'} \exp(g(\theta; x)_{y'}),$$

where the term $\text{LogSumExp}(\cdot)$ is a smooth approximation of the model's largest incorrect logit $\max_{y' \neq y} g(\theta; x)_{y'}$.

Carlini et al. provide a justification for using logit-scaled confidence values instead their unscaled values, or even the cross entropy loss. Notably, it is shown that logit-scaled confidence values provide a more Gaussian distribution. Gaussianity is important because it allows a more efficient parametric modeling of null and alternative distributions $Q_{\text{in}}$ and $Q_{\text{out}}$. In our work, we optimize these parametric distributions by optimizing the canary sample, preserving Gaussianity is equally important to us.

It is shown that in a neural network that produces pre-softmax (i.e. unnormalized) values $g(x)$, logit-scaled confidence $\phi(\frac{p}{1-p})$ where $\phi$ is the logit function, and $p$ are confidence scores such that $p = \text{softmax}(g(x))$. These logit-scaled confidence values can be calculated with higher numerically stability using what Carlini et al. refer to as the "hinge" loss but we prefer to call this function of the trained model $\theta$, and canary sample $(x, y)$ a *measurement function*:

$$f(\theta; x, y) \triangleq g(\theta; x)_y - \text{LogSumExp}_{y'} \, g(\theta; x)_{y'}.$$

By computing the measurement difference between a model trained with and without the canary, we define a computational *privacy loss*:

$$\ell_{\mathrm{priv}}(x, y) = f(\theta_{D \cup \{(x,y)\}}; x, y) - f(\theta_D; x, y).$$

Sampling a privacy canary, therefore, reduces to solving the bilevel problem

$$\max_{(x,y)} \ell_{\mathrm{priv}}(x, y) = f(\theta_{D \cup \{(x,y)\}}; x, y) - f(\theta_D; x, y) \tag{5}$$

$$\text{s.t.} \quad \theta_{D \cup \{(x,y)\}} \in \arg\min_\theta \frac{1}{|D|+1} \sum_{z_i \in D \cup \{(x,y)\}} \mathcal{L}(\theta; z_i), \quad \theta_D \quad \in \arg\min_\theta \frac{1}{|D|} \sum_{z_i \in D} \mathcal{L}(\theta; z_i).$$

**Remark.** To simplify optimization and avoid discrete gradients over the label space, in this work we consider canaries with randomly chosen labels and only optimize the sample $x$. We find this strategy to be sufficient in finding highly distinguishable canaries—measured in TPR@0.1FPR. In our ablations (Section 6.3) we find that combining mislabeling and sample optimization does not improve canary's distinguishability.

**Canary optimization through gradient descent.** With a concretized privacy loss definition in Equation (4), our iterative algorithm to optimize the privacy loss $\ell_{\mathrm{priv}}$ follows in Algorithm 1. The simplicity of the algorithm belies its main difficulty, namely, calculating the canary gradient $\nabla_x \ell_{\mathrm{priv}}(x, y)$.

The gradient step (aka, canary gradient) is $\nabla_x \ell_{\mathrm{priv}}(x) = \nabla_x f(h(\theta_{D \cup \{x\}}; x)) - \nabla_x f(h(\theta_D; x))$. The term $\nabla_x f(h(\theta_{D \cup \{x\}}; x))$ seeks to maximize the privacy measurement $f$. However, this depends on $x$ both in the training set and at inference-time. The

---

**Algorithm 1** Iterative Canary Optimization

**Require:** Initial canary $x$ with label $y$, Training set $D_{\mathrm{train}}$, Canary learning rate $\eta_c$
1: **for** $t \in [T]$ **do**
2:     Train model $\theta^*$ on $D_{\mathrm{train}} \cup \{x\}$
3:     Estimate $\nabla_x \ell_{\mathrm{priv}}(x, y)$
4:     Update the canary: $x = x + \eta_c \nabla_x \ell_{\mathrm{priv}}(x, y)$

---

term $\nabla_x f(h(\theta_D; x))$ tries to minimize the original model's privacy measurement on $x$. Since $\theta_D$ does not have $x$ as a training-time dependency, optimizing this quantity is very similar to finding an **adversarial example**. The core challenge in optimizing the privacy loss in Algorithm 1 is therefore two fold: choosing an appropriate initial canary $x$ and calculation of the gradient $\nabla_x f(h(\theta_{D \cup \{x\}}; x))$.

## 5   OptiFluence: Initialize with Influence, Optimize with Unrolled

In this section, we introduce our canary optimization method, **OptiFluence** which comprises two steps: i) initializing the optimization using influence functions (Section 5.1), and ii) optimization of the canary by efficient differentiation through unrolled model updates in Section 5.2.

### 5.1   IF-Init: Initialization Through Zeroth-order Optimization of Influence

Real-world data distributions often contain rare sub-modes or memorized outliers (Carlini et al., 2019b), which are difficult to generalize from and therefore prone to memorization. We sample our initial canary from such rare points using influence functions (see Section 3), which quantify interactions among training samples. While heuristic, our approach is motivated by the conjecture that samples primarily influenced only by themselves are precisely those rare points: because they cannot be explained by the rest of the training distribution, the network is forced to memorize them.

We define the self-influence metric that we use for this purpose as follows:

**Definition 2** (Normalized Self-Influence). *Assume we have trained a model on the training set $D_{train} = D \cup \{x_i\}$, we define sample $i$'s* normalized self-influence *as its self-influence divided by the largest amount of (cross-)influence every other sample has on $x_i$. Formally,*

$$r_i = \frac{I(x_i; x_i)}{\max_{x_j \in D} I(x_i; x_j)}, \tag{6}$$

*Reminder from Section 3 that $\alpha I(x_i; x_j)$ is the change in sample $x_i$'s loss if we increased sample $x_j$'s weight in training loss by $\alpha$.*

We initialize our canary optimization with the maximizer of normalized self-influence: $x_c = \text{argmax}_{x_i \in D_{\text{train}}} r_i$. Writing the maximum normalized self-influence as

$$\max_{x_i \in D_{\text{train}}} r_i = \max_{x_i \in D_{\text{train}} = D \cup \{x\}} \frac{I(x_i; x_i)}{\max_{x_j \in D} I(x_i; x_j)}, \tag{7}$$

we note the connection to the Neyman-Pearson statistic $\Lambda$ for the membership inference hypothesis test in Equation (3) and (after taking a $\log$) to the objective function in canary optimization problem (5). In maximizing the normalized self-influence we have restricted our measurement function to influence values we calculate over a discrete dataset $D_{\text{train}}$; therefore we are not optimizing over a compact set; and therefore cannot achieve the optima in problem (5). However, combining this pre-selection step to initialize our unrolled gradient optimizer we can indeed go beyond discrete zero-order optimization; and achieve canaries with high TPR at low FPR—reaching 100% TPR@0.1FPR.

## 5.2 UNROLLED-OPT: IMPLICIT DIFFERENTIATION THROUGH UNROLLED MODEL UPDATES

Consider the measurement function $f$ parameterized in terms of the canary input $(x, y)$. We have $f(\theta^*(x); x, y) = g(\theta^*(x); x)_y - \text{LogSumExp}_{y'} g(\theta^*(x); x)_{y'}$, where $\theta^*(x) = \text{argmin}_\theta \ \ell_{\text{train}}(\theta; D_{\text{train}} \cup \{(x, y)\})$ is the trained model parameterized on input $(x, y)$. The privacy loss is

$$\ell_{\text{priv}}(x, y) = f(\theta^*(x); x, y) - f(\theta^*(0); 0, y).$$

Therefore, the second term is constant, and the canary gradient is simply $\frac{d}{dx} f(\theta^*(x); x, y)$. Using the chain rule,

$$\frac{df(\theta^*(x); x, y)}{dx} = \frac{\partial f}{\partial g} \cdot \frac{\partial g}{\partial \theta^*} \cdot \frac{d\theta^*}{dx},$$

where $\frac{\partial f}{\partial g}$ replaces the hinge-loss gradient, and $\frac{\partial g}{\partial \theta^*}$ is the gradient of the canary-class logit, computable via a backward pass on model parameters.

**(a) Full unroll & full backprop**

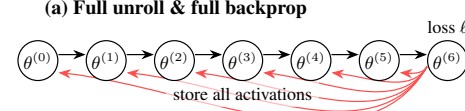

**(b) Rematerialization (checkpointing)**

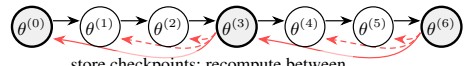

store checkpoints; recompute between

**(c) TBPTT + Rematerialization (window $K$)**

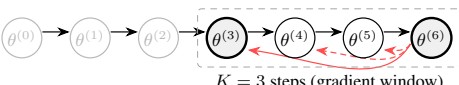

$K = 3$ steps (gradient window)

Figure 2: **Improving memory utilization in a trade-off with time (Rematerialization) and approximation (TBPTT).** Legend: ◯ state, ⬤ stored checkpoint, ◯ truncated, → forward, → backprop, -→ recompute+backprop.

The key issue is the term $\frac{d\theta^*}{dx}$, since $\theta^*(x)$ is implicitly defined. Calculating this gradient requires the implicit function theorem, as in Appendix A.1. Here, instead, we *unroll the gradient updates* of the model after $T$ optimization steps as

$$\theta^* := \theta^{(T)} = \theta^{(0)} - \eta \sum_{t=0}^{T-1} \nabla_\theta \ell\big(\theta^{(t)}; D_{\text{train}} \cup \{(x, y)\}\big).$$

Replacing the training loss $\ell(\theta; D_{\text{train}} \cup \{(x, y)\}) = \frac{1}{n} \sum_i \mathcal{L}(\theta; z_i) + \mathcal{L}(\theta; x, y)$, we have

$$\frac{d\theta^*}{dx} = -\eta \sum_{k=0}^{T-1} \left[ \prod_{j=k+1}^{T-1} (I - \eta H^{(j)}) \right] \nabla_x \nabla_\theta \mathcal{L}\big(\theta^{(k)}; x, y\big), \tag{8}$$

where $H^{(j)}$ is the Hessian of the training loss at step $j$. In practice, we do not calculate Equation (8) explicitly; instead, we rely on automatic differentiation through the computational graph of the unrolled updates.

## 5.3 REMAT+TBPTT: RE-MATERIALIZATION WITH TRUNCATED BACKPROPAGATION

The Jacobian $\nabla_x \nabla_\theta \mathcal{L}\big(\theta^{(k)}; x\big)$ in Equation (8) is the only term that directly depends on $x$ while every other term in the sum depends on $x$ through previous parameter updates, which means the entire computational graph needs to be kept in memory. When differentiating through many training steps, the computational graph grows linearly with the number of updates, leading to prohibitive memory usage. An alternative is to trade memory for extra computation by check-pointing and recomputing parts of the computational graph. This is a well-known strategy in the Hyperparameter Optimization literature known as **Rematerialization** (or, ReMat for short). Using this technique, we trade memory for compute: instead of storing all intermediate states $\{\theta^{(t)}\}_{t=0}^T$, only a subset is kept.

During backpropagation, missing states are recomputed by re-running the forward updates, reducing memory at the cost of additional forward passes. See Figure 2.

**Truncated Backpropagation Through Time (TBPTT)** further addresses scaling by truncating gradient propagation (Williams & Peng, 1990). Rather than computing $\nabla_x \ell = \sum_{t=0}^{T-1} \frac{\partial \ell}{\partial \theta^{(T)}} \prod_{j=t+1}^{T} \frac{\partial \theta^{(j)}}{\partial \theta^{(j-1)}} \frac{\partial \theta^{(t)}}{\partial x}$, TBPTT approximates it with a shorter horizon $\nabla_x \ell \approx \sum_{t=T-K}^{T-1} \frac{\partial \ell}{\partial \theta^{(T)}} \prod_{j=t+1}^{T} \frac{\partial \theta^{(j)}}{\partial \theta^{(j-1)}} \frac{\partial \theta^{(t)}}{\partial x}$ where $K \ll T$. This reduces both compute and memory, while still providing informative gradient signals.

Together, rematerialization and TBPTT enable unrolled optimization of privacy canaries over long training runs without exhausting GPU memory.

## 6 EVALUATION

In this section, after presenting our empirical setup in Section 6.1, we evaluate the effectiveness of our canaries in Section 6.2. We present a thorough ablation study in Section 6.3 to validate our design choices for OptiFluence.

**IF-Opt: First-Order Optimization of Influence Functions.** One of our baselines in this section is a novel method based on derivatives of influence functions, called IF-Opt. In this approach, we present a new method to calculate the gradient canary using the model's inverse-Hessian-vector products (IVHFs). This formulation of the canary gradient allows us to benefit from the advances in calculation of IVHFs of large neural networks (e.g. EK-FAC approximations) which are primarily studied in the Training Data Attribution literature. In the interest of presentation and given the superior performance of ReMat+TBPTT, we decided the delegate the detailed discussion of IF-Opt to Appendix A.

### 6.1 EXPERIMENT SETUP

**Canary Optimization.** We evaluate OptiFluence on two datasets: CIFAR-10 and MNIST. We note that these are standard datasets used for privacy auditing due to the fact that computational efficiency is a key factor—often one requires training hundreds (and often thousands) of models using these datasets. On MNIST, we train small two-layer MLPs, and consider both Unrolled-Opt and IF-Opt for canary optimization. Due to the larger computational cost of CIFAR-10, we start with hyperlightspeedbench (Balsam, 2023) CNNs for IF-Opt. Then, since Unrolled-Opt is more computationally expensive due to the unrolling steps, we apply ReMat+TBPTT on the optimizer to enable training on ResNet-9, 18, 50 (He et al., 2016) and WideResNet16-4 (Zagoruyko & Komodakis, 2016). For MNIST we optimize canaries in all settings and audit using 20k victim models per canary. For CIFAR10, we audit using 128 victim models. Unless otherwise stated, we run three trials for each method setup.

**Auditing Procedure.** We consider two auditing setups: *traditional DP auditing* (Jagielski et al., 2020) and an *efficient auditing heuristic* (Aerni et al., 2024). In the traditional audit setup, we train many "victim" models for every single canary while varying the canary's membership. Whenever this is computationally infeasible, we instead use the heuristic approach of Aerni et al. (2024), which reuses victim models to audit multiple canaries simultaneously. In both cases, we use the LiRA membership inference attack (Carlini et al., 2022) with the "hinge" score, and follow contemporary practices by reporting the attack's true positive rate (TPR) at a fixed low false positive rate (FPR) (see Figure 7 in Appendix C for ROC curves). For training victim and shadow models, we use a training oracle that randomly samples a training dataset from a fixed known base dataset $D$.[1] Following Carlini et al. (2022), we ensure that each canary is a member of the training data in exactly half of the models. Unless stated otherwise, we always use the same training procedure for canary optimization, victim models, and shadow models.

**Baselines.** For baselines, we include in-distribution (ID) examples (i.e. randomly sampled from the training dataset), mislabeled examples (Nasr et al., 2021; Steinke et al., 2023; Aerni et al., 2024), and adversarial examples (Nasr et al., 2021). We defer the exact training and optimization details to Appendix B.

---

[1]We compare this to a training oracle that includes the full $D$ in the training data in Appendix C.

## 6.2 VALIDATION

Table 1: Canary effectiveness measured as TPR in % at 0.1% FPR on MNIST and CIFAR-10.

| Methods | ID | Mislabeled | Adversarial | **OptiFluence** |
|---|---|---|---|---|
| MNIST | $0.12 \pm 0.03$ | $0.40 \pm 0.12$ | $0.45 \pm 0.15$ | $\mathbf{99.83 \pm 0.14}$ |
| CIFAR-10 | $0.24 \pm 0.06$ | $0.33 \pm 0.17$ | $36.98 \pm 25.79$ | $\mathbf{99.48 \pm 0.43}$ |
| CIFAR-100 | - | - | - | $\mathbf{100.0 \pm 0.0}$ |

Table 2: Measured wall-clock time (seconds) and peak VRAM usage (GB) represented as time/VRAM usage for the canary initialization step and optimization step of OptiFluence and the baselines on MNIST, CIFAR10 and CIFAR100.

| | Initialization Step | | Optimizaiton Step | |
|---|---|---|---|---|
| | Wall-clock Time (s) | Peak VRAM Usage (GB) | Wall-clock Time (s) | Peak VRAM Usage (GB) |
| MNIST | $2,641$ | 2.3 | 32 | 79 |
| CIFAR10 | $7,456$ | 38 | $1,128$ | 80 |
| CIFAR100 | $7,458$ | 38 | $1,292$ | 80 |

**OptiFluence produces canaries with nearly optimal TPR@0.1FPR.** As shown in Table 1, OptiFluence achieves almost perfect canary detectability on both datasets: 99.83% $\pm$ 0.14% on MNIST and 99.48% $\pm$ 0.43% on CIFAR10 TPR at 0.1% FPR. Canaries generated by OptiFluence demonstrated significant performance enhancement compared to baselines. On MNIST, it is up to 831.9× more effective than in-distribution canaries, 249.6× more effective than mislabeled canaries, and 221.8× more effective than adversarial canaries. For CIFAR-10, our canary is 414.5× more effective compared to in-distribution canaries, 301.5× more effective compared to mislabeled canaries. Although compared to adversarial, we are 2.69× more effective, given the high variance 25.79% of it, our method clearly improved over it. We refer readers to Figure 1 and Appendix C for the qualitative results of each method.

**OptiFluence scales with larger datasets.** We compare the wall–clock time and peak VRAM usage of OptiFluence across all datasets. As shown in Table 2, OptiFluence scales from the small MNIST dataset to CIFAR10 and CIFAR100, while maintaining near perfect performance. The optimization step peaks at 80GB of VRAM usage on a single H100 GPU, which can be further reduced by decreasing the TBPTT window size $K$ without degrading performance. Adjusting other hyper-parameters, such as batch size or learning rate, can also reduce the runtime, which may introduce some trade–offs in accuracy. Note that CIFAR100 is more complex than CIFAR10 and therefore requires more training epochs, which impacts computational cost. Nevertheless, our method still achieves a significantly high TPR@0.1%FPR, and with a configuration that maintains comparable computational costs as CIFAR10. The canary initialization step requires longer computation, but it is performed only once, and IF-Init ensembles 20 models for robustness, contributing to the increased runtime; the per–model runtime is approximately 373s. Safely reducing the ensemble size to 5 offers a substantial runtime reduction with minimal effect on final canary selection performance.

**Optimized canaries transfer between architectures.** Transferability of input-space canaries is important because it enables efficient *third-party audits*: auditors can require model builders to include these samples in their training sets (possibly with a zero-knowledge proof of sampling (Shamsabadi et al., 2024)). OptiFluence-optimized canaries achieves strong transferability: our best CIFAR10 canaries trained on ResNet-9, achieve nearly perfect TPR@0.1FPR scores for ResNet-18, ResNet-50 and WideResNet16-4—despite differences in architecture, training regime (e.g. learning rate scheduling for the larger models, etc.) In Appendix C.4, we present transferability results for Unrolled-Opt baseline over different-width MLPs on MNIST and similarly observe a strong transferability despite architectural differences.

**DP-SGD Auditing.** Our canaries are architecture and training-regime-agnostic, as such we can use them for auditing models trained with DP-SGD. However, we note no input-space canary (including ours) provides tight auditing bounds for DP-SGD, as the adver-

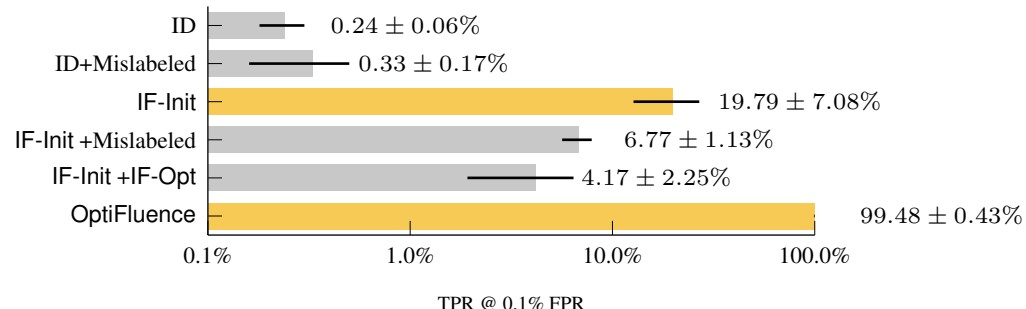

Figure 3: We compare different design selections of OptiFluence at each stage of the pipeline, on CIFAR10, yellow indicates our choice at each stage.

sary model simply does not have access to the sampling procedure, nor can it manipulate model updates—a requirement for tight auditing of DP-SGD (Nasr et al., 2021).

We train 3 ResNet9 models on CIFAR10 with privacy parameter $\varepsilon \in \{1, 2, 4, 6, 8\}$, clipping norm of $1.0$ and use a standard Renyi-DP accountant (`dp-accounting`) while ensuring that we include our most sensitive canary in the training set. Table 3 demonstrates the results. As expected, attacks success improves with larger privacy budgets; but is significantly diminished compared to non-privatized models (99% to 1.6% for the most private model). Despite this, even against the most private model, OptiFluence achieves attack success an order of magnitude better than the baseline canary of mislabeling in-distribution data points (1.6% vs. 0.33%) from Figure 3.

| $\varepsilon$ | TPR@0.1FPR |
|---|---|
| 0.5 | 0.0% |
| 1 | 1.6% |
| 2 | 1.6% |
| 6 | 4.7% |
| 8 | 6.2% |

Table 3: **Auditing DP-SGD with OptiFluence-optimized canaries.** We report the Global threshold TPR (Aerni et al., 2024).

### 6.3 ABLATIONS

We perform ablation studies on CIFAR10 to explain our design choice at each stage of the OptiFluence pipeline as shown in Figure 1. Each primitive selected improved performance of the previous stage, evolution trajectory shown in Figure 3.

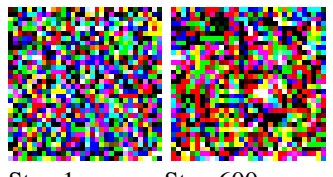

Step 1            Step 600

Figure 4: **A TBPTT-trained canary for CIFAR10.** Distinct patterns emerge but the sample is not visually in-distribution. This achieves 18.8% TPR@0.1FPR.

**Proper initialization is essential when using truncated backpropagation.** Rematerialization trade-offs time for memory but the unrolled gradients remain exact. TBPTT however skips earlier steps of training which introduces approximation errors in the unrolled gradient. Figure 4 demonstrate a trained canary with ReMat+TBPTT for CIFAR10. Despite showing distinct patterns compared to the initial steps of training, the final canary does not resemble a CIFAR10 image—this is in contrast with Figure 6 that resembles MNIST digits. In other words, **the canary is not strictly in-distribution**. Note that compared to baselines (in-distribution, mislabeled, and adversarial) IF-Init achieves a higher attack success (at 19.8% TPR@0.1FPR). As a result, with proper initialization we can do much better.

**Combined with canary optimization, mislabeling is not effective.** From Figure 3, we observe that for canaries selected with IF-Init, adding mislabeling after such zeroth-order initialization contributes little to attack success rate. In contrast, when we apply mislabeling to IF-Init canaries, performance drops by approximately 2.92×. We see a similar trend for IF-Opt initialized with IF-Init under mislabeling. Although for in-distribution examples, mislabeling yields a slight improvement (approximately 1.38×), mislabeling appears to shift the data toward a different mode that resembles out-of-distribution. While such a perturbation can be stronger than the in-distribution baseline, IF-Init already targets tail samples and outperforms both baselines, adding mislabeling pushes them further to the OOD direction and thus degrades performance.

**Unrolled optimization produces stronger canaries.** Finally, to evaluate different optimization options for canaries in typical empirical privacy auditing, we discuss IF-Opt, Unrolled-Opt, and ReMat+TBPTT. Note that ReMat+TBPTT is a more memory-efficient implementation to fit Unrolled-Opt in memory for ResNet9 on CIFAR10.

As shown in Figure 3, IF-Init +IF-Opt with no mislabeling is $17.37\times$ more effective than the ID baseline. However, IF-Init +IF-Opt underperformed IF-Init. We argue the reason is methodological, IF-Opt optimizes over influence functions, which relies on the EK-FAC approximation of the inverse Hessian for the Inverse Hessian-vector products Grosse et al. (2023). EK-FAC assumes layer independence, although the approximation is theoretically upper bounded for deeper networks with skip-connections, such as ResNets, the approximate error remains unknown. Furthermore, influence functions also assume convergence and convexity Hammoudeh & Lowd (2024). Combined, these assumptions introduce variance and noise, as evidenced in Figure 3 where IF-Init +IF-Opt has a standard error as high of 2.25%.

Unrolled-Opt evaluated the training data influence by differentiating through the entire training trajectory, sidestepping the convergence assumption. Since it does not apply the Implicit-Function Theorem, there are no similar curvature approximations as in IF-Opt and Unrolled-Opt yields an exact gradient of our measurement (hinge loss) with respect to the training data (canary). Optimizing over these exact gradients produces much stronger result with lower variance. On CIFAR10, for computational feasibility of ResNet9, we introduced ReMat in which the gradients remain exact, however, ReMat+TBPTT introduces approximation by gradient truncation to further improve computation efficiency. Empirically, this trade-off did not show harm to canary detectability in our privacy auditing setting.

All of the steps combined, culminated in our final method OptiFluence, which achieved $99.5\pm0.43\%$ TPR@0.1%FPR.

## 7 CONCLUSIONS

We introduced OptiFluence, a principled framework for privacy auditing that formulates canary design as a bilevel optimization problem. By combining influence-based preselection with memory-efficient unrolled optimization, our method systematically constructs canaries that maximize their detectability under membership inference attacks. This approach addresses the limitations of prior heuristic strategies—such as mislabeled or out-of-distribution samples—that may underestimate privacy leakage and yield inconsistent auditing signals.

Our empirical results on MNIST and CIFAR-10 demonstrate that optimized canaries achieve up to $415\times$ higher detectability than heuristic baselines, while also transferring effectively across model architectures. These findings show that privacy auditing can move beyond ad hoc constructions toward a rigorous and scalable methodology.

Looking forward, we believe that optimized canaries can serve as standardized auditing primitives for both practitioners and regulators. They not only provide tighter empirical lower bounds on privacy leakage, but also establish a foundation for auditing frameworks that are model- and domain-agnostic. Future work should extend these ideas to larger datasets, federated and distributed settings, and the auditing of differentially private training at scale. Ultimately, our results suggest that principled canary optimization is a key step toward reliable and reproducible privacy audits in modern machine learning.

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

## A  IF-OPT: FIRST-ORDER OPTIMIZATION WITH INFLUENCE FUNCTIONS

### A.1  BACKGROUND

Influence functions are a classical technique from robust statistics that were made scalable by Koh & Liang (2017) to explain large ML model's predictions by means of identifying the "most influential" training samples for a given prediction.

Assume we have trained the optimal model parameters $\theta^*$ on a training set $D_{\text{train}} = \{z_i \mid z_i = (x_i, y_i)\}$, where $x_i, y_i$ are input feature and labels. Using a *per-sample loss function* $\mathcal{L}(\theta; z_i)$ we have the classical ERM optimization problem $\theta^* = \text{argmin}_{\theta \in \mathbb{R}^d} \frac{1}{N} \sum_{i=1}^{N} \mathcal{L}(\theta; z_i)$.

The classical influence setting assumes the optimizer $\theta^*$ exists and are unique. We want to understand the effect of adding (or removing) a new training example $z_c$ to the training set. We can parametrize the optimizer using the weight $\alpha$ of this sample: $\theta^*(\alpha) = \text{argmin} \frac{1}{N} \sum_i \mathcal{L}(\theta; z_i) + \alpha \mathcal{L}(\theta; z_c)$ We define the *influence* of $z_c$ on $\theta^*$ via its first-order Taylor approximation of the response function at $\alpha = 0$. Given sufficient regularity, we can use the Implicit Function Theorem (Krantz & Parks, 2002) to calculate the influence as:

$$I_{\theta^*}(z_c) \triangleq \frac{d\theta^*}{d\alpha}\bigg|_{\alpha=0} = -H^{-1} \nabla_\theta \mathcal{L}(\theta^*; z_c), \tag{9}$$

where $H = \nabla_\theta^2 \ell(\theta^*; D_{\text{train}})$ is the Hessian of the optimized model parameters on the training set. Using a first-order approximation of Equation (9) and setting $\alpha = \frac{1}{N}$, we can estimate the influence of changing sample $z_c$'s weight on the final model parameters with $\theta^*(\alpha) - \theta^* \approx \alpha I_{\theta^*}(z_c) = -\alpha H^{-1} \nabla_\theta \mathcal{L}(\theta^*; z_c)$.

Note that $\theta^*(\alpha) - \theta^* \in \mathbb{R}^D$, i.e., the influence lives in the (possibly very large) weight-space. To make influence more interpretable, it is customary to define a *measurement* function $f$ and instead ask: how does changing sample $z_c$'s weight changes the measurement, i.e., $f(\theta^*(\alpha)) - f(\theta^*)$?

In the data attribution literature, a good measurement function is often the validation loss or the logits for the query example $z_c$. Using the chain rule, calculating the influence on measurement function $f$ is straightforward:

$$f(\theta^*(\alpha)) - f(\theta^*) \approx \alpha I_f(z_c) = -\alpha \nabla_\theta f(\theta^*)^\top H^{-1} \nabla_\theta \mathcal{L}(\theta^*; z_c) \tag{10}$$

where $I_f(\cdot)$ is the influence on measurement $f$.

### A.2  CANARY OPTIMIZATION USING INFLUENCE FUNCTIONS

Returning to our original goal of calculating the gradient canary $\nabla_x \ell_{\text{priv}}(x, y)$, our reformulation of the privacy loss (Equation (4) in Section 4) appears as self-influence in Equation (10) which gives us a functional tool to maximize the privacy loss $\ell_{\text{priv}}(x, y)$.

Concretely, we use Koh & Liang (2017, Section 2.2). It is shown that the impact of perturbing a training sample $x$ to $x + \Delta x$ on a test sample $x_c$ measurement (e.g. loss) $f(h(\theta_{D \cup \{x\}}; x_c))$ is approximated by $dI_{f, \theta_{D \cup \{x\}}}^\top \Delta x$ where

$$dI_{f, \theta_{D \cup \{x\}}}(x, x_c) = -\nabla_\theta f(h(\theta_{D \cup \{x\}}; x_c))^\top H_\theta^{-1} \nabla_x \nabla_\theta \mathcal{L}(h(\theta_{D \cup \{x\}}; x)) \tag{11}$$

This is maximized when we set $\Delta x = \kappa dI_{f, \theta_{D \cup \{x\}}}(x, x_c)$ for some positive scalar $\kappa$. We choose the test point to be $x_c = x$ to match the quantity we seek to optimize. Therefore, we have

$$\nabla_x f(h(\theta_{D \cup \{x\}}; x)) = dI_{f, \theta_{D \cup \{x\}}}(x, x)$$

Note that $\nabla_\theta \mathcal{L}(h(\theta_{D \cup \{x\}}; x))$ is a Jacobian (w.r.t. input $x$) calculated over the canary's per-sample gradient (w.r.t model parameters $\theta$). Therefore, the gradient is of size $O(n^2 d)$ where $n^2$ is the number of inputs, and $d$ is the number of model parameters. To make this calculation viable for larger models, we: i) calculate the inverse-Hessian-vector product (IVHF) $H^{-1} \nabla_\theta \mathcal{L}(\theta^*; x)$ using the EK-FAC approximations; and b) implement the Jacobian as a pure function allowing the use of automatic gradient (autograd). We offer more implementation details in Appendix A.3.

### A.3 INFLUENCE FUNCTION DETAILS

We rely on a `functorch`-based version of `kronfluence` to calculate our influence functions. The importance of `functorch` is to generate influence as a pure function; enabling the calculation of the Jacobian in Equation (11). In this section, we briefly review concepts used in this library to measure influence functions over non-converged and non-convex models. Our treatment in this section is largely based on Grosse et al. (2023).

**Proximal Bregman Response Function.** In Appendix A.1 we discussed the the limitations of influence functions. Bae et al. (2022) show that calculating influence functions on non-converged or non-convex models amounts to training the model not with the ERM loss but with a modified objective known as the proximal Bregman objective (PBO). The resulting response function is known as the proximal Bregman response function (PBRF):

$$\theta^s(\alpha) = \arg\min_{\theta \in \mathbb{R}^D} \frac{1}{N} \sum_{i=1}^{N} D_{\mathcal{L}_i}\left(h\left(\theta; x_i\right), h\left(\theta^s; x_i\right)\right) + \frac{\lambda}{2} \left\|\theta - \theta^s\right\|^2 + \alpha \mathcal{L}\left(\theta; z_c\right), \quad (12)$$

where $\theta^s$ are the final (but not necessarily converged) model weights $\hat{y}_i = h(\theta; x_i)$ is the prediction of the model on sample $x_i$. The $D_{\mathcal{L}_i}(\cdot, \cdot)$ denotes the Bregman divergence for the output space loss function: $D_{\mathcal{L}_i}(\hat{y}, \hat{y}^s) = \mathcal{L}_y(\hat{y}, y_i) - \mathcal{L}_y(\hat{y}^s, y_i) - \nabla_{\hat{y}} \mathcal{L}_y(\hat{y}^s, y_i)^\top (\hat{y} - \hat{y}^s)$.

Applying the implicit function theorem as before on the new response function, we can calculate the influence on the PBRF:

$$I_{\theta^s}\left(z_m\right) = \left.\frac{d\theta^s}{d\alpha}\right|_{\alpha=0} = -(G + \lambda \mathbf{I})^{-1} \nabla_\theta \mathcal{L}\left(z_c, \theta^s\right). \quad (13)$$

Here, $G$ denotes the Gauss-Newton Hessian (GNH), defined as $G = \mathbb{E}[J^\top H\hat{y}J]$, where $J = d\hat{y}/d\theta$ represents the Jacobian of the model outputs with respect to the parameters, and $H\hat{y}$ is the Hessian of the loss function with respect to the outputs. The expectation is taken over the empirical data distribution.

Crucially, the projected Bayesian objective (PBO) remains well-defined even in the case of over-parameterized or partially trained neural networks. Unlike the full Hessian $\mathbf{H}$, the matrix $\mathbf{G}$ is guaranteed to be positive semidefinite, and the regularized version $\mathbf{G} + \lambda \mathbf{I}$ is strictly positive definite for any $\lambda > 0$. Prior studies (Bae et al., 2022; Grosse et al., 2023) have therefore adopted the damped Gauss-Newton Hessian $\mathbf{G} + \lambda \mathbf{I}$ as a surrogate in influence function computations—an approach we also follow in this work.

**EK-FAC for fast inverse-Hessian-vector products (IHVPs).** EK-FAC treats the curvature of each linear or convolutional layer as a Kronecker product of two much smaller empirical covariance matrices. If the layer's Jacobian $J \in \mathbb{R}^{B \times P}$ (batch $B$, parameters $P$) is written as $J = (A \otimes S)$, then the Gauss-Newton/Hessian block is approximated by

$$H \approx G_{\text{EK-FAC}} = \underbrace{\left(\frac{1}{B} A^\top A\right)}_{\Phi \in \mathbb{R}^{n \times n}} \otimes \underbrace{\left(\frac{1}{B} S^\top S\right)}_{\Psi \in \mathbb{R}^{m \times m}},$$

so $H^{-1}$ factorizes as $(\Phi^{-1} \otimes \Psi^{-1})$. In a practical implementation, we:

1. **Collect statistics.** After a forward–backward pass, accumulate
$$\Phi = \frac{1}{B} A^\top A, \qquad \Psi = \frac{1}{B} S^\top S;$$
where exponential moving averages keep the factors up-to-date at low cost.

2. **Add damping.** Ensure positive-definiteness and numerical stability:
$$\Phi \leftarrow \Phi + \lambda I, \quad \Psi \leftarrow \Psi + \lambda I.$$

3. **Invert factors.** (once per update) Small sizes ($n, m \leq$ few hundred) let us do an exact Cholesky/SVD solve $\Phi^{-1}$, $\Psi^{-1}$.

4. **Form IHVP for any vector (v).** Reshape the layer slice of $v$ into matrix form $V \in \mathbb{R}^{n \times m}$. Apply the Kronecker inverse by two cheap solves:
$$\text{vec}\left(\Phi^{-1} V \Psi^{-1}\right) = \left(\Phi^{-1} \otimes \Psi^{-1}\right) v \approx H^{-1} v.$$

To calculate the IHVP for the whole-network, we repeat steps 1–4 for every layer and concatenate the results; no conjugate-gradient iterations are needed and each matrix–vector product costs $O(n^3 + m^3 + nm)$, independent of parameter count.

# B EXPERIMENT DETAILS

We release a preliminary version of our code in the supplementary material as a zipped file and will publish a cleaned version with the camera-ready version of this paper.

## B.1 OPTIMIZATION AND TRAINING HYPERPARAMETERS

If not mentioned otherwise, all experiments use the exact same model architecture and training procedure when optimizing canaries and when training models for auditing. For all MNIST experiments, we use a small MLP with two layers and a hidden dimension of 20 features. We train those MLPs using SGD with a learning rate of 0.1 and momentum 0.9 with batch size 128 for 10 epochs. On CIFAR-10, we use hyperlightspeedbench (Balsam, 2023) CNNs with default training hyperparameters if not mentioned otherwise, for ReMat+TBPTT, we use ResNet9.

## B.2 CANARY INITIALIZATION

We used influence functions (Kronfluence) to compute influence scores in Equation (7). On MNIST, we used the same architecture as for canary optimization and model training. On CIFAR-10, we used the HyperLightSpeedBench CNN for efficiency. To improve robustness, we averaged our metric over an ensemble of 20 independently trained models before sorting and selecting the canaries. For IF-Init, we selected three canaries from the top five ranked by the metric for each of the three trials in every experiment.

## B.3 CANARY OPTIMIZATION IMPLEMENTATION DETAILS

**Unrolled optimization.** We reduce variance in unrolled optimization by calculating gradients over multiple models simultaneously. Concretely, at each step, we train two "IN" models on a dataset that contains the canary and two "OUT" models on a dataset without the canary. Next, we calculate the mean hinge score of all "IN" models and all "OUT" models. We then maximize the difference using gradient ascent with a learning rate of 1.0 and momentum 0.9 for 300 steps. Every initial canary is a uniformly random image.

**Influence optimization.** In contrast to unrolled optimization, influence optimization uses an existing training sample (that will be replaced) as the initial canary. Optimization then repeatedly trains a model and maximizes the canary's self-influence on that model using gradient ascent. We estimate each canary's self-influence using the full base dataset (which is in the threat model of having access to the data distribution). For MNIST models, we use a canary learning rate of 0.001, no momentum, and 50 update steps; for CIFAR-10, we use a canary learning rate of 0.001, momentum 0.9, and 100 update steps.

In both cases, we use early stopping to combat high variance in the optimization procedure. At each optimization step, we calculate the canary's hinge score on a model that was trained on the canary to the model that was trained on the initial canary from the base dataset; we use the canary with the largest gap in scores.

### B.4   AUDIT SETUP

---

**Algorithm 2** LiRA Membership Inference Audit (for one sample $x$)

---

**Require:** Base dataset sampler $\mathcal{O}$, candidate sample $x$, shadow counts $N_{\text{in}}$, $N_{\text{out}}$ (typically equal), victim model $h_v$, score function $s(h, x)$ (e.g., hinged scaled logit), evaluation grid of thresholds $\mathcal{T}$

---

1: **Shadow training**:
2: **for** $i = 1, \ldots, N_{\text{in}}$ **do**
3:     Draw $D_i \sim \mathcal{O}$ and set $D_i^{\text{in}} = D_i \cup \{x\}$
4:     Train shadow model $h_i^{\text{in}}$ on $D_i^{\text{in}}$
5: **for** $j = 1, \ldots, N_{\text{out}}$ **do**
6:     Draw $D_j' \sim \mathcal{O}$ and set $D_j^{\text{out}} = D_j'$
7:     Train shadow model $h_j^{\text{out}}$ on $D_j^{\text{out}}$
8: **Score aggregation**:
9: $S_{\text{in}} \leftarrow \left\{ s(h_i^{\text{in}}, x) \mid i = 1..N_{\text{in}} \right\}$
10: $S_{\text{out}} \leftarrow \left\{ s(h_j^{\text{out}}, x) \mid j = 1..N_{\text{out}} \right\}$
11: **Fit score models (Gaussian LiRA)**:
12: Fit $\mathcal{N}_{\text{in}} = \mathcal{N}(\mu_{\text{in}}, \sigma_{\text{in}}^2)$ to $S_{\text{in}}$
13: Fit $\mathcal{N}_{\text{out}} = \mathcal{N}(\mu_{\text{out}}, \sigma_{\text{out}}^2)$ to $S_{\text{out}}$
14: **Victim evaluation**:
15: Obtain victim score $z_v \leftarrow s(h_v, x)$
16: Compute log-likelihood ratio

$$\text{LLR}(x) \;=\; \log p_{\mathcal{N}_{\text{in}}}(z_v) - \log p_{\mathcal{N}_{\text{out}}}(z_v)$$

17: **Audit metric**:
18: **for** threshold $\tau \in \mathcal{T}$ **do**
19:     Predict "member" if $\text{LLR}(x) \geq \tau$, else "non-member"
20: Compute ROC and report TPR at desired FPR (e.g., 0.1%)

---

For all results, we perform LiRA-style (Carlini et al., 2022) membership inference with 128 shadow models and the hinge score. That is, we first train shadow models while ensuring that each sample is a member in exactly half of them. Then, for every canary, we fit a Gaussian distribution on all member and non-member scores, and use their log-likelihood ratio on a victim models' score as the attack statistic. Similarly, we ensure that every sample is a member of exactly half of the victim models.[2]

We calculate ROC scores over predictions from victim models by using membership as the label, and we again ensure that each canary is a member in exactly half of the models' training data. Whenever we audit multiple canaries in parallel, we concatenate all their attack scores (following Aerni et al. (2024)).

As mentioned in Section 6.1, we consider two auditing setups: *traditional DP auditing* and an *efficient auditing heuristic*. Given the aforementioned membership inference procedure, those two setups only differ in the number of victim and shadow models. Concretely, we use 20k victim models for a single canary (resulting in 20k predictions) in the traditional setup, and the heuristic setup uses 512 victim models with 32 canaries (resulting in around 16k predictions). The heuristic setup allows us to obtain auditing results for a broader set of settings at the cost of slightly underestimated worst-case privacy leakage (Aerni et al., 2024).

### B.5   BASELINE CANARIES

We use three types of baseline canaries: in-distribution samples, mislabeled samples, and adversarial examples. For in-distribution baselines, we simply use the original image and label of the sample that would be replaced with a canary. For mislabeled samples, we use the same images, but flip their label to a different one selected uniformly at random. Lastly, we optimize adversarial examples using

---

[2]In Appendix C.3, we only enforce this for canaries, and include all samples of the base dataset in all models' training data.

the FGSM attack (Goodfellow et al., 2015) with an epsilon of $0.3$ and the $\ell_\infty$-norm. To improve robustness of the resulting adversarial examples, we maximize the minimum loss over 8 models.

## C  ADDITIONAL EXPERIMENTS

### C.1  CANARIES AT DIFFERENT METRIC QUANTILES

In this section, we show a distribution of our metrics score and the canary samples and the attack results at 1% (our main result that produced the best canary examples), 5%, 10% and 25%.

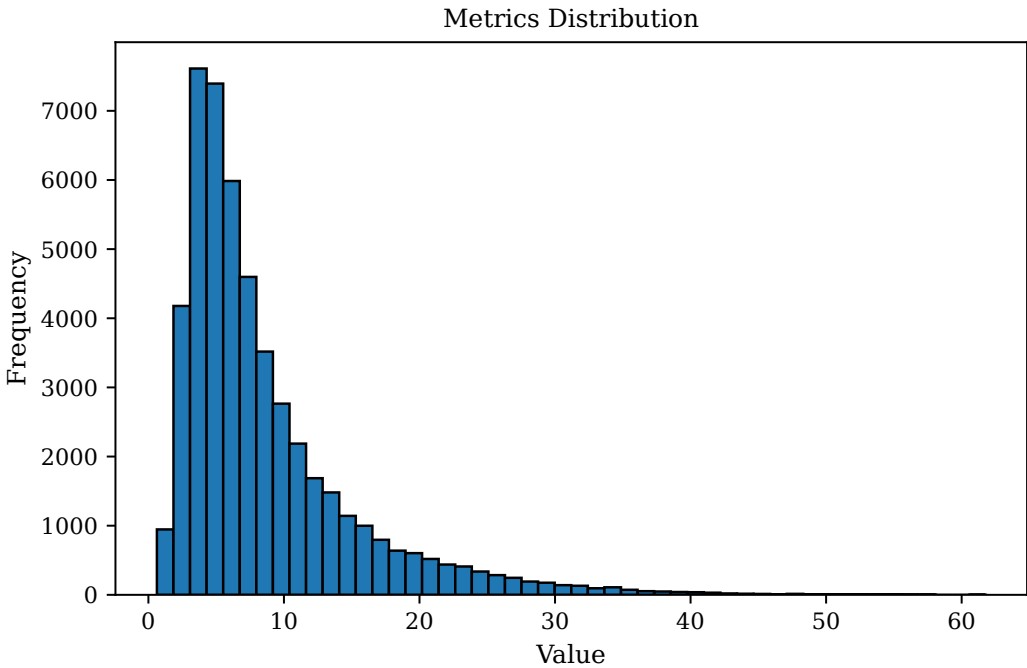

Figure 5: We show that the metrics of the training data follows a long-tail distribution on CIFAR10 dataset.

### C.2  QUALITATIVE EXAMPLES AND ROC CURVES

Figure 6 displays the optimized canaries from our main experiment (Table 1). On MNIST, the resulting canaries exhibit distinct patterns. For example, one unrolled optimized canary for MNIST (bottom left corner) resembles a negative of a two. In contrast, on CIFAR-10, the resulting canaries still largely resemble their initialization from real data. However, for one influence-optimized MNIST canary, the optimization procedure did not introduce any perceptible changes. The canary's ROC curve highlights that this is likely due to suboptimal optimization; the corresponding line in Figure 7 ("MNIST Influence-Optimized", green dotted) shows a consistently lower TPR.

### C.3  EFFECTS OF SUB-SAMPLING THE BASE DATASET

In our main experiments, we always sample a random subset of a given base dataset when training models (for optimizing canaries and for auditing). This corresponds to an adversary that knows the training data distribution, but not necessarily the full non-canary training data. Here, we consider a stronger adversary that knows the full base dataset.

As shown in Figure 8, the difference between the two threat models is negligible. Concretely, we use the same settings as Table 1, but we always include the full base dataset in every model's training data—only varying the membership of the canaries. Due to the computational cost, we only use a single seed.

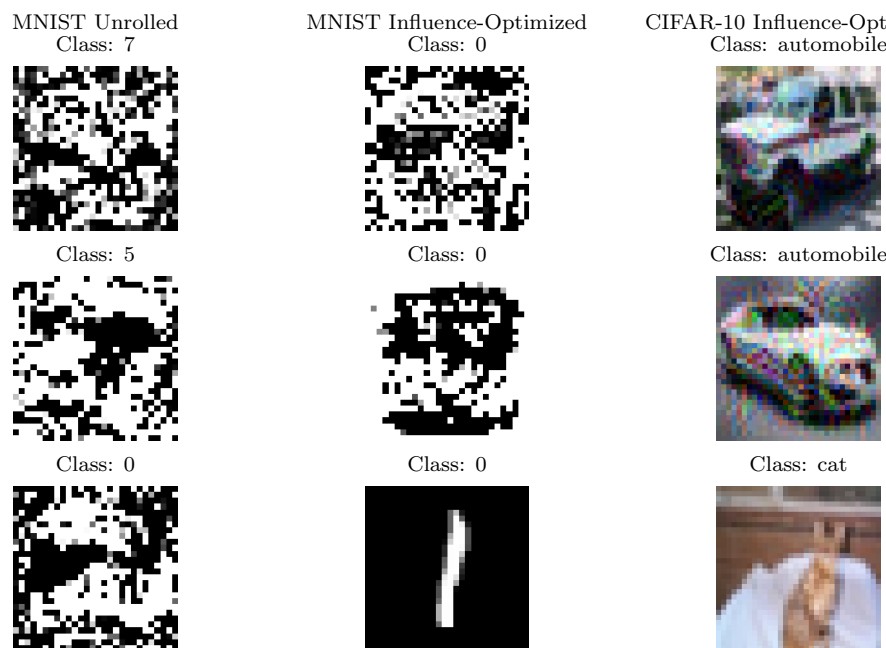

Figure 6: **Optimized canaries exhibit distinct patterns.** We show the canaries from Table 1. On MNIST, canary optimization yields results that often exhibit particular structure, resembling a superposition of features from multiple classes. On CIFAR-10, effects are more subtle.

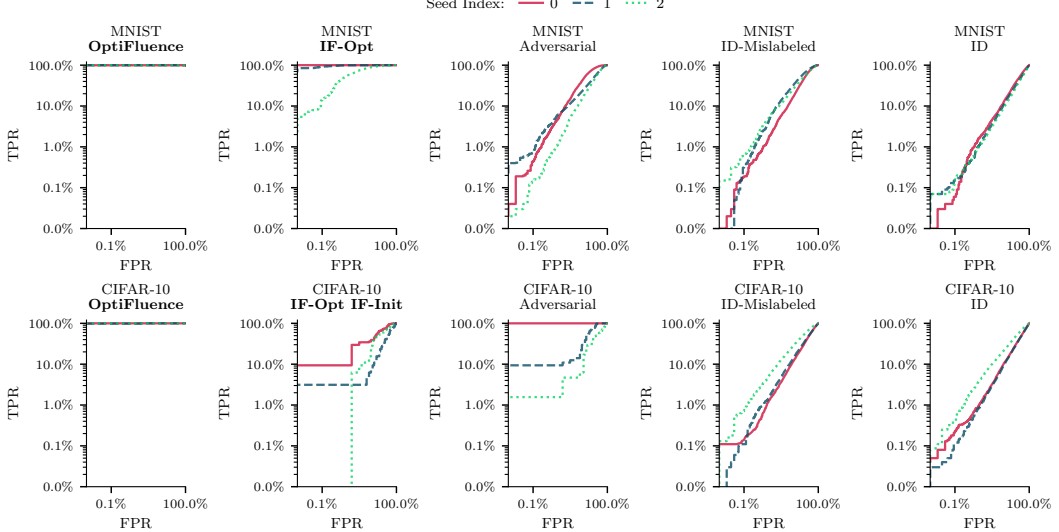

Figure 7: **ROC curves of optimized canaries.** We show all ROC curves of the canaries in Table 1. Different lines correspond to different seeds. Note that both the x-axis and y-axis are on a logarithmic scale.

## C.4 TRANSFERABILITY: STUDY ON MLPs

An important property of optimized canaries is transferability. Since generating canaries can be costly, one would hope that canaries optimized on a small model are still strong canaries for larger models. We hence optimize canaries for MNIST on the same MLPs as before, but then use those canaries to audit MLPs with larger widths. This corresponds to a threat model with a weaker adversary, where the adversary does not have perfect knowledge of the training procedure. Due to the large computational cost, we use the heuristic auditing scheme with 512 victim and 128 shadow models.

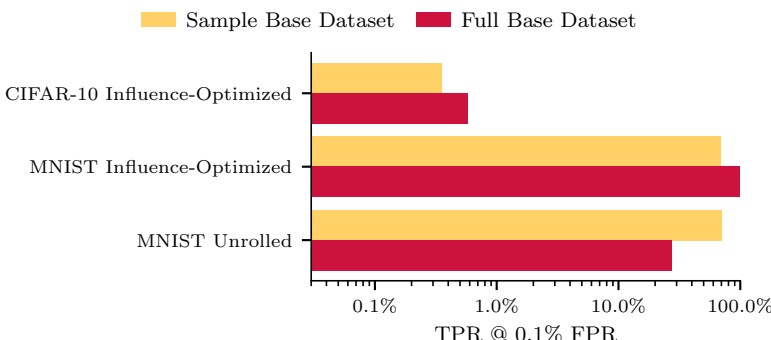

Figure 8: **Using the full base dataset has negligible impact on optimized canaries.** We compare our standard setting, which samples a random half of the base dataset (yellow), to a setup that always includes the full base dataset in the training data (red). Using the full base dataset has little impact on the auditing power of all optimized canary types.

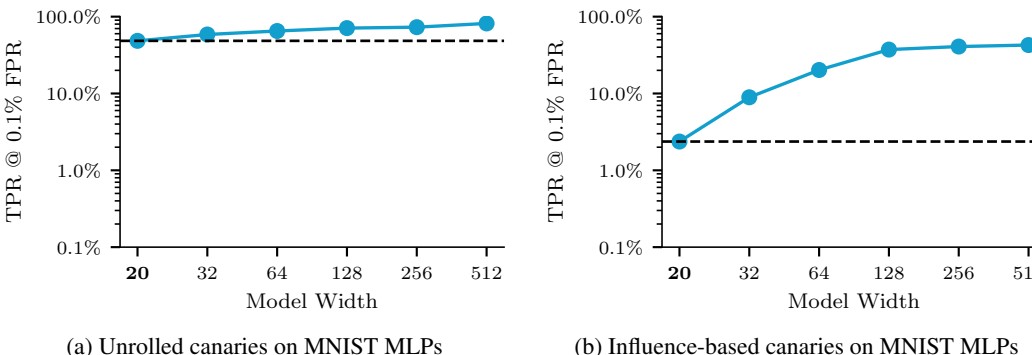

(a) Unrolled canaries on MNIST MLPs  (b) Influence-based canaries on MNIST MLPs

Figure 9: **Optimized canaries transfer between model architectures.** We optimize a canary for a width-20 MLP and use it to audit models of larger widths. Canaries transfer well to larger models, exhibiting similar or even higher TPR at low FPR. We use the efficient audit heuristic with 512 victim and 128 shadow models.

As seen in Figure 9, canaries optimized on the smallest models are still strong canaries for larger models. In fact, we find that the TPR at 0.1% FPR even increases as model size increases. This hints that optimized canaries strongly exploit the memorization capabilties of a given model architecture, thereby generalizing beyond the specific model used for optimization. This property, in combination with recent work on efficient unrolling of SGD (Bae et al., 2024; Engstrom et al., 2025), can enable more effective auditing of much larger models such as LLMs.

### C.5 DISTINGUISHABILITY

**Optimization makes canaries more distinguishable.** We investigate how our optimization procedures make canaries more distinguishable. Concretely, we optimize a canary using both unrolled and influence-based optimization on an MLP for MNIST. We then take the canary after one step of optimization, after some intermediate steps, and at the end. For each, we fit 64 models with the canary in their training data and 64 models without. The results in Figure 10 show that our optimization procedure primarily affects models not trained on a canary; decreasing the LiRA hinge score for non-member models while retaining the score for member models.

### C.6 ADDITIONAL TRANSFERABILITY RESULTS

Figure 9 shows that canaries optimized on one model transfer to larger models. We present additional results in the following: we compare TPR at low FPR of optimized canaries to mislabeled samples and include results on influence-optimized canaries on CIFAR-10. There, we scale the base widths

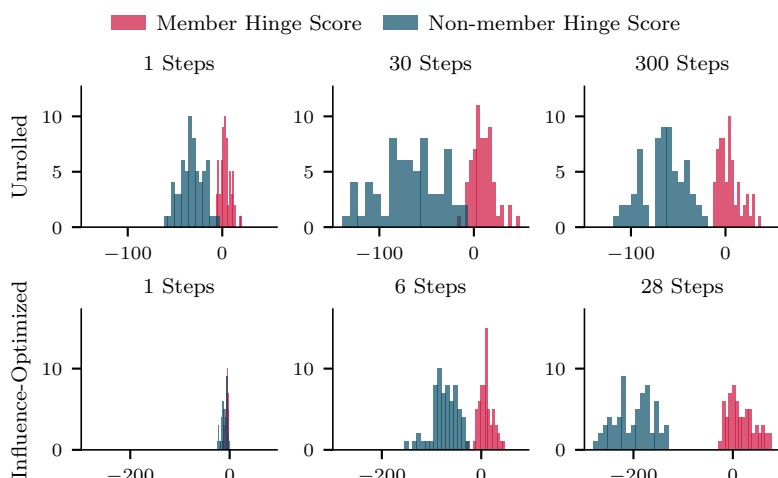

Figure 10: **Optimizing canaries reduces non-member scores.** We use intermediate canaries when using unrolled (top) and influence-based (bottom) optimization for a small MNIST MLP and train 64 models each with and without the canaries. Our optimization procedures primarily work by reducing the scores of non-member models.

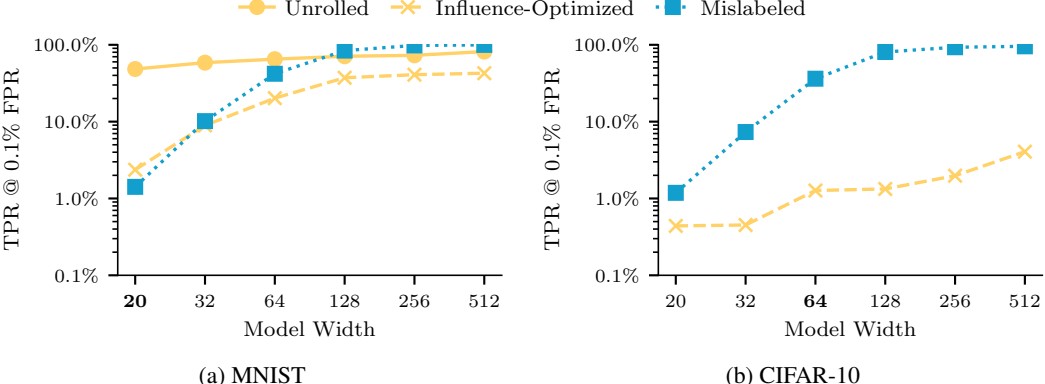

Figure 11: **Additional transferability results.** We extend the results of Figure 9 by additionally investigating the transferability of influence-optimized canaries on CIFAR-10 and baseline mislabeled samples. On MNIST (a), canaries from unrolled optimization achieve high TPR at low FPR even when the model width is too small to memorize mislabeled samples. For CIFAR-10 (b), influence-optimized canaries also exhibit favorable transferability, but do not yet match the TPR of mislabeled samples at much larger models.

(number of filters) of the HLB CNN; we optimize canaries at a base width of 64, and consider the same range of widths as for the MLPs on MNIST. The rest of the setup is the same as in Figure 9.

As highlighted in Figure 11, influence-optimized canaries and mislabeled samples exhibit a similar scaling curve. However, Figure 11a shows that unrolled canaries can achieve a very high TPR at low FPR even for very small widths, where the models' capacities are too small to memorize mislabeled samples. Figure 11b shows that even on CIFAR-10, canaries optimized on a smaller model scale to larger models, although they do not match the TPR of mislabeled samples yet. We conjecture that additional engineering effort in the influence optimization procedure can close this gap.

### C.7 IMPACT OF THE TRUNCATION PARAMETER $k$ IN TRUNCATED BACKPROPAGATION THROUGH TIME

In Figure 12, we show the canary loss during optimization for three different choices of k: 2, 4, and untruncated for canaries trained on our MLP model on MNIST. The latter is using our Unrolled-Opt baseline where no truncation occurs.

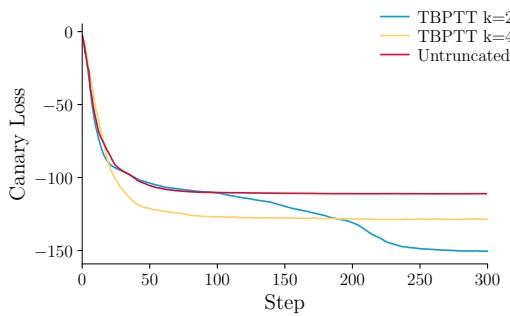

Figure 12: **Canary loss comparison between values truncation parameter** $k$. In practice, all three canaries achieve nearly perfect detectability (99% TRR@0.1FPR scores).

Despite these difference, we did not observe a meaningful impact to tuning $k$ in practice. Lower $k$ does not in fact impact the TPR@lowFPR scores of our attacks. As even canaries trained with $k = 2$ remain nearly perfectly detectable.

## D THE ORTHOGONALITY CONDITION: AN EXTENSIVE COMPARISON WITH MADDOCK ET AL. (2023) AND NASR ET AL. (2023)

In Maddock et al., Algorithm 1 we note the optimization is over the canary sample $z$ which happens in the input-space through minimization of the following canary loss:

$$\min_{z \in \mathbb{R}^d} \mathcal{L}(z) = \sum_i \langle u_i, C \cdot \nabla_\theta \ell(z) \rangle^2 + \max(C - \|\nabla_\theta \ell(z)\|, 0)^2 \qquad \text{(Line 5)}$$

Note that however, calculating the above loss, requires a "pool of clients" that send model updates $u_i$ (Line 2); therefore crafting the canary $z$ requires a federated set-up with a pool of model updates are available (which is not a requirement in our setting).

**Weight-space vs. input-space.** We can write the above loss $\mathcal{L}(z_t)$ re-factorized in terms of a canary gradient $u_c \triangleq \nabla_\theta \ell(z_t)$ :

$$\min_{u_c \in \Theta} \tilde{\mathcal{L}}(u_c) = \sum_i \langle u_i, C \cdot u_c \rangle^2 + \max(C - \|u_c\|, 0)^2 .$$

While we certainly *can* optimize canaries in the input-space $\mathbb{R}^d$, we can do so directly in the space of model parameters $\Theta$. Consequently, the effective threat model in Maddock et al. is weight-space gradients. This is supported by the pipeline in their Figure 2 where we clearly see that the adversary releases the update canary $u_c$ and not the canary sample $z$.

Nasr et al. (2023, Algorithm 3) implements a canary loss that seeks to "align" the canary gradient with the average in-distribution gradient $\vec{g}_{\text{dist}}$:

$$\min_{(x,y)} l_{\text{adv}}(x,y) = \left| \frac{\nabla l(\theta, (x,y)) \cdot \vec{g}_{\text{dist}}}{|\nabla l(\theta, (x,y))| |\vec{g}_{\text{dist}}|} \right| \text{ where } \vec{g}_{\text{dist}} = \frac{1}{|D|} \sum_{(x_i, y_i) \in D} \nabla l(\theta, (x_i, y_i)) \qquad (14)$$

Unlike CANIFE, the threat model here indeed is the release of input-space canaries. We share results using our implementation of Algorithm 3 on MNIST.

**Empirical Results using Nasr et al. (2023, Algorithm 3).** We optimize the objective (14) down to $\ell_{\text{adv}} \leq 0.0001$. We then evaluate the resulting canaries using the same empirical setup as in Section 6.1 but using 5,000 shadow models:

- Initializing the in-distribution sample (following Line 4 of the algorithm) we achieve, 0.2% TPR@0.1FPR.

- Initializing from a canary sampled uniformly at random, and optimizing using Algorthim 3, we achieve 7.4% FPR@0.1FPR.

These results are close to in-distribution and adversarial example baselines for MNIST in Table 2, respectively. What these baseline share is the fact that the canary gradients are not being shaped by the model training dynamics. *The average in-distribution gradient in-distribution gradient $\overrightarrow{g}_{dist}$ is essentially a constant.*

Comparing this to our bi-level objective formulation in Eq. 5 (reproduced here for convenience):

$$\max_{(x,y)} \ell_{\text{priv}}(x, y) = f(\theta_{D \cup \{(x,y)\}}; x, y) - f(\theta_D; x, y) \tag{re.5}$$

$$\text{s.t.} \quad \theta_{D \cup \{(x,y)\}} \in \arg\min_{\theta} \frac{1}{|D|+1} \sum_{z_i \in D \cup \{(x,y)\}} \mathcal{L}(\theta; z_i), \quad \theta_D \in \arg\min_{\theta} \frac{1}{|D|} \sum_{z_i \in D} \mathcal{L}(\theta; z_i),$$

We note that the first constraint depend on the canary that is being put in the training set, therefore, a change in the canary $(x, y)$ lead to changes in the resulting model $\theta_{D \cup \{(x,y)\}}$ that needs to be taken into consideration in the unrolled loss. Nasr's $\ell_{\text{adv}}$ does not take the impact of the inclusion of the canary in the training procedure directly as we do, but rather approximates it by optimizing the proxy objective $\ell_{\text{adv}}$ .

Nasr et al. justify their choice of the proxy objective empirically, noting in Section B.2 "using the dot product between the privatized gradient and the canary gradient is a sufficient metric for auditing DP-SGD." For the sake of the presentation, let us call this the "orthogonality condition."

To the best of our knowledge Maddock et al. 2023, first derived and justified the orthogonality condition. Notably, in Appendix A, authors connect their objective to the likelihood ratio test and derive the condition under the assumption that "model update follow Gaussian distribution $\mathcal{N}(\mu, \Sigma)$ . The sum of $k$ model updates then either follows $\mathcal{N}(k\mu, k\Sigma)$ (without the canary) or $\mathcal{N}(k\mu + \nabla\ell(z), k\Sigma)$ (with the canary), recalling that $u_c \propto \nabla\ell(z)$ ." Under this assumption, Maddock et al. derive the likelihood ratio test between the null $p_0$ and alternative $p_1$ distributions:

> " In particular for the centers of the Gaussian with and without the canary, $u \in \{k\mu, k\mu + \nabla\ell(z)\}$ ,
>
> $$\log\left(\frac{p_1(u)}{p_0(u)}\right) = \pm\frac{1}{2}\nabla\ell(z)^T(k\Sigma)^{-1}\nabla\ell(z).$$
>
> Maximizing this term will thus help separate the two Gaussians. **However, doing this directly is infeasible as it requires to form and invert the full covariance matrix $\Sigma$ in very high dimensions.** Instead, we propose to minimize $z \mapsto \left(\nabla\ell(z)^T\right)\Sigma(\nabla\ell(z))$ as it is tractable and can be done with SGD. Note that for sample model updates $\{u_i\}$ we can estimate the (uncentered) covariance matrix as $\frac{1}{n}\sum_i u_i u_i^T$ and thus
>
> $$\left(\nabla\ell(z)^T\right)\Sigma(\nabla\ell(z)) \approx \frac{1}{n}\sum \nabla\ell(z)^T \left(u_i u_i^T\right)\nabla\ell(z) = \frac{1}{n}\sum \langle u_i, \nabla\ell(z)\rangle^2 .$$
>
> "

We see that **the key approximation that leads to the orthogonality condition is the inverse-product-Hessian (IVHP) $\nabla\ell(z)^T(k\Sigma)^{-1}\nabla\ell(z)$ which Maddock et. al consider infeasible to do with SGD. But this IVHP calculation is exactly what we do efficiently** using influence function in Appendix A (for the IF-OPT baseline—see Eq. 10) and improve upon using unrolled gradients in the main matter for OptiFluence!

## STATEMENT ON LLM USE

We have used LLMs for the purposes of a) re-writing and paraphrasing text in the paper; and b) coding and implementation of some of the techniques.

