# OpenReview forum: "OptiFluence: Scalable and Principled Design of Privacy Canaries"
_ICLR.cc/2026/Conference — Submitted to ICLR 2026_

### Official Review · Reviewer_t7cL · 2025-10-28

**Soundness:** 2
**Presentation:** 3
**Contribution:** 2
**Rating:** 4
**Confidence:** 5

**Summary:**

This paper proposes OptiFluence, a bilevel optimization framework for designing privacy canaries that maximize their detectability under membership inference attacks. The method integrates influence-based pre-selection and unrolled sample optimization with memory-efficient techniques. Experiments on MNIST and CIFAR-10 demonstrate strong detectability.

**Strengths:**

* The paper tackles an important problem in empirical privacy auditing, providing a more principled alternative to heuristic canary constructions.

* The bilevel optimization formulation is elegant and connects privacy auditing with influence functions and gradient unrolling.

**Weaknesses:**

* **Incomplete component description in the abstract.**
The abstract claims that OptiFluence consists of three components but describes only two. Please correct this inconsistency.

* **Lack of discussion on the relationship to adversarial examples.**
As noted around L228–L229 and Algorithm 1, the optimization process resembles adversarial example generation.
The paper should explicitly discuss how the proposed canary differs from conventional adversarial samples, conceptually and in objective formulation, and clarify why existing adversarial methods cannot directly be used to generate canaries. Experimental comparisons between OptiFluence and adversarial samples are also needed.

* **Missing threat-model specification.**
The current presentation lacks a clear statement of the auditor’s capability—whether auditing assumes black-box, gray-box, or white-box access to the model.
A formal threat model is essential to contextualize the results and interpret the claimed transferability.

**Questions:**

* **Unclear explanation of transferability.**
While Section 6.2 claims strong cross-architecture transfer, the supporting evidence in Appendix C.4 (two MLPs on MNIST) is insufficient to substantiate general transferability.
The authors should explain why optimized canaries can transfer between architectures (e.g., shared feature space, loss geometry) and include more diverse models or quantitative analyses.

* **Potential performance degradation.**
Algorithm 1 suggests that the canary is iteratively updated during model training. This may interfere with model convergence or degrade performance.
The paper should report whether incorporating such optimization affects model accuracy or training stability.

* **Relation to privacy–robustness trade-off.**
Given that OptiFluence’s optimization resembles adversarial training, it would be valuable to evaluate or at least discuss the potential trade-off between privacy auditing effectiveness and robustness, as widely documented in the literature [1–3].
An experiment showing canary auditing after adversarial training would significantly strengthen the paper.

[1] Zhang, Z., Zhang, L. Y., Zheng, X., Abbasi, B. H., & Hu, S. (2022). Evaluating membership inference through adversarial robustness. The Computer Journal, 65(11), 2969-2978.
[2] Lyu, L. et al. (2022). Privacy and robustness in federated learning: Attacks and defenses. IEEE TNNLS, 35(7), 8726-8746.
[3] He, F., Fu, S., Wang, B., & Tao, D. (2020). Robustness, privacy, and generalization of adversarial training. arXiv preprint arXiv:2012.13573.

---

> ### Author Response · Authors · 2025-11-24
>
> We thank the reviewer for their detailed feedback. Since the points raised in the weaknesses and the questions can have a significant overlap, we will be answering them jointly. We use numbered W(eakness) and Q(questions) to refer to the comments. We kindly invite the reviewer to consult the **updated manuscript** while considering our responses, where *additions are marked in blue*, and *fixes are marked in red*.
>
> > W1: Incomplete component description in the abstract
>
> Thank you for pointing out this inconsistency. We have fixed the issue in the updated manuscript.
>
> > W2/Q3: The paper should explicitly discuss how the proposed canary differs from conventional adversarial samples, conceptually and in objective formulation, and clarify why existing adversarial methods cannot directly be used to generate canaries.  Experimental comparisons between OptiFluence and adversarial samples are also needed.
>
> We believe the manuscript already addresses both requests. First, Table 1 in Section 6.2 presents a baseline of using Adversarial examples, which at 0.45% TPR@0.1FPR is only marginally better than Mislabeled examples at 0.40%.
>
> Second, in the aforementioned section (Lines 228-229), in the loss objective that we derive for optimizing the hypothesis test statistic (Eq. 3), the gradient resembling that of an adversarial example appears with a negative sign. That is, based on the derivation, we in fact do not want canaries to look like adversarial examples. Motivating this through years of research in adversarial training (whereby we train with adv. examples for robustness) is illuminating: since canaries are, by definition, going to be trained on, we want them to remain sensitive.
>
> > Q3. Relation to privacy–robustness trade-off.
> An experiment showing canary auditing after adversarial training would significantly strengthen the paper.
>
> Note that, as shown in Algorithm 1, indeed the model is always trained with the canary in the training set (Line 2). Therefore, we indeed do adversarial training for all results. As discussed,  we seek to "robustify" our canaries to be trained on.
>
> > W3: Missing threat-model specification.
> The current presentation lacks a clear statement of the auditor’s capability—whether auditing assumes black-box, gray-box, or white-box access to the model.
>
> We agree that a threat model would be helpful; however, **we do not think that the black vs. white-box dichotomy is the right level of granularity for our approach.** The fact that our canaries are transferable makes this clear: an auditor does not need access to the model architecture to optimize a good canary; so this can be considered a black-box. However, we have shown how much choosing an influential sample for the initialization of canary optimization makes a difference. This requires at least some knowledge about the auditee's data distribution. But does that mean auditing requires white-box access to the auditee's data? We believe not. This level of knowledge is assumed for any realistic auditor.
>
> Since the black/white-box characterization limits auditing to the box (aka, the algorithm), it fails to consider who does the auditing, which is a much more important consideration. **Privacy auditing with canaries enables third-party audits; while algorithmic audits (such as "Tight-", or "One-run" audits) often require first-party access, which is often characterized as white-box.**
>
> > Q1: Unclear explanation of transferability.
> While Section 6.2 claims strong cross-architecture transfer, the supporting evidence in Appendix C.4 (two MLPs on MNIST) is insufficient to substantiate general transferability.
>
> In Line 424-426, we already share that we have trained multiple model architectures (ResNets 18, 50, and WideResNet) with canaries trained on ResNet9 and achieved nearly perfect (99% TPR@0.1FPR). So our transferability results are not only on MLPs.

---

> ### Author Response · Authors · 2025-11-24
>
> > Q1. Continued: The authors should explain why optimized canaries can transfer between architectures (e.g., shared feature space, loss geometry)
>
> A theoretical study of this phenomenon is outside the scope of the current paper. We can provide the following observational explanation of this phenomenon:
>
> > All models (i.e., hypothesis classes) seek to learn the same concept from the data. A transferable canary indicates that the notion of a canary is not a function of the hypothesis class, but rather the concept class itself. For example, for digit classification, we know that a 2 and a 7 are reasonably close to each other; and, written in bad handwriting, one can be mistaken for the other. Therefore, a good canary can be an image that can reasonably be classified as either 2 or a 7 by even a human, an entirely different learner!
>
> We like to note, however, that the space of canaries is much larger than the above example. But the above should be sufficient to show why transferability of canaries would make sense in the first place.
>
> Please also see our response to Reviewer_VkF8.
>
> > Q2. Potential performance degradation.
> Algorithm 1 suggests that the canary is iteratively updated during model training. This may interfere with model convergence or degrade performance.  The paper should report whether incorporating such optimization affects model accuracy or training stability.
>
> Indeed, in prior work, injecting samples (like poisons) comes at a trade-off with model performance. However, please note that, unlike prior work, **we inject exactly one sample as the canary to stay maximally compliant with the DP definition.** This means that **the contribution of the single sample error to the loss is 1/|Size of Training Set|; therefore, we do not observe any noticeable degradation in the accuracy of the model.** This is the reason we do not report model accuracy consistently. There is not much to report.
>
> This is, of course, understandable because performance is an average metric while privacy is a worst-case one. **We optimize the worst-case (an "outlier" sample), which does not affect the average considerably.**
>
> We thank the reviewer for their insightful commentary and hope to have answered their questions. If so, we would appreciate it if they increased their score. We welcome any follow-up questions.

---

### Official Review · Reviewer_5uPC · 2025-10-28

**Soundness:** 2
**Presentation:** 3
**Contribution:** 2
**Rating:** 4
**Confidence:** 3

**Summary:**

This work introduces OptiFluence, an optimization-based framework for conducting membership inference attacks (MIAs) against differentially private (DP) models. Unlike prior works in the central setting that often rely on sample-based canaries, OptiFluence formulates the attack as a gradient-based optimization problem in which the attacker crafts a synthetic sample to maximize their influence on the model output or loss. The authors argue that this exposes vulnerabilities in DP training not captured by existing stochastic or heuristic attacks. Experiments on standard benchmarks show that OptiFluence achieves higher attack success rates under certain scenarios.

**Strengths:**

- The optimization-based formulation is well described and the approach of combining influence functions to intitialize an optimized canary is novel.
- The attack is evaluated across a good range of datasets and privacy budgets.

**Weaknesses:**

- The exact contribution over prior work that utilizes optimized canaries in the white-box or federated setting is unclear (see below).
- Aspects of the presentation related to the specific names of proposed/baseline methods could be improved to make the experiments more easily readable (see questions below).
- The optimization process seems to be computationally expensive. Although the authors propose an approximation denoted, ReMat+TBPTT, no experimental results demonstrate its effectiveness or runtime benefits. This makes it seem less practical than simpler alternatives (e.g., one-run or random canary methods).

**Questions:**

1. What exactly  is IF-OPT? It doesn’t seem to be clearly defined in the experiments.
2. Could the authors provide runtime/overhead comparisons against the baselines to substantiate claims of scalability? The approximation approach ReMat+TBPTT is proposed but there seem to be no results that show why you should use it over the fully unrolled updates. There are no statements in the paper about how long this canary optimization process takes? Is this attack really practical for auditing?
3. Given the higher computational cost, what practical advantage does OptiFluence offer over one-run or random canary insertion attacks?
4. I would have liked to have seen a clearer ablation on the impact of using the influence function intialization vs. standard canary optimization (w/o this IF initialization) which appears to be missing from Figure 3?
5. How does OptiFluence fundamentally differ from the optimization-based canaries proposed by [1,2]?

[1] Nasr, Milad, et al. "Tight auditing of differentially private machine learning." 32nd USENIX Security Symposium (USENIX Security 23). 2023.

[2] Maddock, Samuel, Alexandre Sablayrolles, and Pierre Stock. "Canife: Crafting canaries for empirical privacy measurement in federated learning." arXiv preprint arXiv:2210.02912 (2022).

---

> ### Author Response · Authors · 2025-11-24
>
> We thank the reviewer for their detailed feedback. Since the points raised in the weaknesses and the questions can have a significant overlap, we will be answering them jointly. We use numbered W(eakness) and Q(questions) to refer to the comments. We kindly invite the reviewer to consult the **updated manuscript** while considering our responses, where *additions are marked in blue*, and *fixes are marked in red*.
>
> > W1/Q5: The exact contribution over prior work that utilizes optimized canaries in the white-box or federated setting is unclear (see below). How does OptiFluence fundamentally differ from the optimization-based canaries proposed by [1,2]?
>
> Maddock et al. 2022's algorithm CANIFE produces optimized canaries in the federated learning setup, which means the canaries are in the parameter space. Our canaries are in the input (sample) space (i.e, model weights vs. pixels for a vision canary). Unfortunately, the two types of canaries cannot be used interchangeably, given the difference in deployment context. We cite Maddock et al. 2022 in Line 107.
>
> In Nasr et al. 2023, tight auditing is possible for "canary gradients," which are again canaries in the parameter space, and not the input space. The closest canary discussed in Nasr et al. 2023 to us is the Black-box Auditing (Algorithm 1), where canaries used are mislabeled examples (thus heuristically chosen). We discuss Nasr et al. 2023 work in Line 40.
>
> [1] Nasr, Milad, et al. "Tight auditing of differentially private machine learning." 32nd USENIX Security Symposium (USENIX Security 23). 2023.
>
> [2] Maddock, Samuel, Alexandre Sablayrolles, and Pierre Stock. "Canife: Crafting canaries for empirical privacy measurement in federated learning." arXiv preprint arXiv:2210.02912 (2022).
>
> > W2/Q1:  What exactly is IF-OPT? It doesn’t seem to be clearly defined in the experiments.
>
> IF-OPT is Influence-Optimized canaries: this is a baseline explained in detail in Appendix A. The approach involves using gradients of influence functions (aka, first-order derivatives of influence functions).
>
> To address your question, in our updated manuscript, we have expanded our existing short description of this baseline in Line 341 in the paragraph titled "IF-Opt: First-Order Optimization of Influence Functions."
>
> > W3/Q3. The optimization process seems to be computationally expensive. Although the authors propose an approximation denoted ReMat+TBPTT, no experimental results demonstrate its effectiveness or runtime benefits.
>
> In Figure 3, we do compare against "random" (which we take to be heuristic) canary methods, such as mislabeling or taking a random in-distribution (ID) sample to be the canary.
>
> One-run method is a privacy auditing technique that takes canaries (many of them) as input (see Line 1 Alg.1 in Steinke et al. [1]). Our contribution is to optimize canaries to be used in privacy audits, so **canaries are the output of our method**. Since we are not contributing a new privacy audit (we use Aerni et. al 2024 as our privacy auditing framework), one run is not a comparable baseline.
>
> [1] Steinke, Thomas, Milad Nasr, and Matthew Jagielski. 2023. “Privacy Auditing with One (1) Training Run.” *Advances in Neural Information Processing Systems* 36 (December): 49268–49280.
>
> > Q2. Could the authors provide runtime/overhead comparisons against the baselines to substantiate claims of scalability?
>
> To answer the reviewer’s comment regarding empirical evidence for the efficiency of our method, we have updated Section 6.2 to report the requested metrics in Table 2 and added a paragraph explaining why our method scales effectively from a small dataset such as MNIST to a more complex dataset like CIFAR-100. We have also updated Table 1, which now shows near-perfect detectability on CIFAR-100 as well.
>
> > Q2. Continued: The approximation approach ReMat+TBPTT is proposed, but there seem to be no results that show why you should use it over the fully unrolled updates.
>
> We do, in fact, motivate the necessity of ReMat+TBPTT at the start of Section 5.3. The (exact) unrolled gradients of Section 5.2 (the Unrolled-OPT baseline) are so memory-intensive that one could not fit the computational graph for any reasonable ResNet  CIFAR10 model in memory. With  ReMat+TBPTT,  however, we can do so for models as large as ResNet50, WideResNet, as well as models trained on more complex datasets such as CIFAR100 (which have a 10x larger final layer compared to CIFAR10). See our response to Reviewer VkF8's W3.

---

> ### Author Response · Authors · 2025-11-24
>
> > Q2.Continued: There are no statements in the paper about how long this canary optimization process takes. Is this attack really practical for auditing?
>
> Empirically we observe that **once the optimization fits in memory, canary optimization takes seconds (for MNIST) or minutes (for CIFAR10) to complete (see Table 2).** By far the most time consuming part of an audit process for us is the training and attacking of the shadow models; which as we have mentioned, we adopt from Aerni 2024 and thus is not a contribution of our work at all. Therefore, canary optimization is not a bottleneck.
>
> > Q4. I would have liked to have seen a clearer ablation on the impact of using the influence function initialization vs. standard canary optimization (w/o this IF initialization), which appears to be missing from Figure 3.
>
> We believe the ablation you are looking for is shown in Figure 1, where we show both an in-distribution sample (car, lower left) that achieves 2.4% TPR@0.1FPR, and after optimization (lower right), achieves 5.2% TPR@0.1FPR. Figure 3 is purposefully used to do a sequential ablation such that improvement using each subsequent choice over the previous one is clear.
>
> We thank the reviewer for their insightful commentary and hope to have answered their questions. If so, we would appreciate it if they increased their score. We welcome any follow-up questions.

---

> > ### Comment · Reviewer_5uPC · 2025-11-27
> >
> > I would like to thank the authors for their detailed rebuttal and revised version of their paper.
> >
> > * For W1/Q5, please correct me if I am wrong, but both referenced works do study input-space canaries? In particular, Nasr et al. consider adversarial examples and their own input-space canary construction (Algorithm 3), see also Appendix C.3. I believe this should be made more explicit in the main paper discussion.
> >
> > * For W2/Q1, thank you for the clarification. The corresponding changes in the main paper make this much clearer.
> >
> > * For Q2, I appreciate the extended response and the new experiments benchmarking the canary optimization runtime.
> >
> > * For Q4, I think I understand your point via Figure 1, which partially addresses my concern for CIFAR-10. However, I would prefer to see this studied more concretely across additional datasets to better illustrate the impact of the IF initialization.

---

> > > ### Author Response · Authors · 2025-11-28
> > >
> > > We are pleased to see that we have answered reviewers' concerns regarding W2/Q1, Q2, and Q4.  We answer the follow-up questions.
> > >
> > > >For W1/Q5, please correct me if I am wrong, but both referenced works do study input-space canaries? In particular, Nasr et al. consider adversarial examples and their own input-space canary construction (Algorithm 3), see also Appendix C.3. I believe this should be made more explicit in the main paper discussion.
> > >
> > > We thank you for the follow-up discussion. We think you have a point here, and we now understand how our wording may confuse the reader regarding the difference with prior work in terms of optimization space (input-space vs. weight-space) vs. application domain (federated learning, first-party auditing, etc.). **Below, we provide an extensive technical discussion and a new empirical result to answer your question definitively.** We have also updated the manuscript again to add a **new Appendix D** with the content of this response and added appropriate references to it in the Related Work section in Line 110.
> > >
> > > We will tackle each work separately to position our technical contribution more clearly with respect to Maddock et al 2023 and Nasr et al. 2023:
> > >
> > > **Maddock et. al 2023's CANIFE**
> > >
> > > In CANIFE's Algorithm 1, the optimization is indeed over the canary sample $z$, which happens in the input space through minimization of the following canary loss:
> > > $$\min_{z \in \mathbb R^d}\mathcal{L}\left(z\right) = \sum_i\left\langle u_i, C \cdot \nabla_\theta \ell\left(z\right)\right\rangle^2+\max \left(C-\left\|\nabla_\theta \ell\left(z\right)\right\|, 0\right)^2 \tag{Line 5}$$
> > >
> > > Note that, however,  calculating the above loss requires a "pool of clients" that send model updates  $u_i$ (Line 2); **therefore, crafting the canary $z$ requires a federated set-up with a pool of model updates available (which is not a requirement in our setting).**
> > >
> > > Regarding weight-space vs. input-space: note that we can write the above loss $\mathcal{L}\left(z_t\right)$ re-factorized  in terms of a canary gradient $u_c \triangleq \nabla_\theta \ell\left(z_t\right)$ :
> > > $$\min_{u_c \in \Theta} \mathcal{\tilde L}(u_c) = \sum_i\left\langle u_i, C \cdot u_c \right\rangle^2+\max \left(C-\left\|u_c\right\|, 0\right)^2$$
> > >
> > > Therefore, while one certainly *can* optimize canaries in the input-space $\mathbb ^d$, they can do so directly in the space of model parameters $\Theta$. Therefore, **the effective threat model here is weight-space gradients. This is supported by the pipeline diagram in Figure 2 where we clearly see that the adversary releases the update canary $u_c$ and not the canary sample $z$.**

---

> > > ### Author Response · Authors · 2025-11-28
> > >
> > > > For Q4, I think I understand your point via Figure 1, which partially addresses my concern for CIFAR-10. However, I would prefer to see this studied more concretely across additional datasets to better illustrate the impact of the IF initialization.
> > >
> > > We understand. To respond to your request we ran more experiments on MNIST, our aggregated results follow:
> > >
> > > | Dataset | Method | Result |
> > > |---|---|---|
> > > | MNIST | IF-Init + Unrolled-Opt | 0.99827 ± 0.00142 |
> > > | MNIST | ID-Init + Unrolled-Opt | 0.70407 ± 0.23573 |
> > > | CIFAR10 | IF-Init + ReMat Unrolled | 0.99479 ± 0.00425 |
> > > | CIFAR10 | ID-Init + ReMat Unrolled | 0.36458 ± 0.25951 |
> > >
> > > These results makes the contribution of initialization clear. **We observe a consistent improvement in attack success using influence functions (IF-Init) compared to ID initialization (picking an In-Distribution sample at random).** However, the gap between the ID- vs. IF-Init is much larger for CIFAR10 compared to MNIST. We attribute this to the fact that we have to use approximate canary gradients (ReMat) for the CIFAR10 model, which makes proper initialization ever more essential to the success of the canary optimization (as discussed in Section 6.3: Proper initialization is essential...).
> > >
> > >
> > >
> > > Thank you for the insightful questions. We understand that due to the ongoing circumstances, you are not allowed to respond to our answers. We nevertheless hope to have answered all your concerns.

---

> ### Author Response · Authors · 2025-11-28
>
> (continued)
>
> **Nasr et. al 2023**
>
> Nasr et al. Algorithm 3 implements a canary loss that seeks to "align" the canary gradient with the average in-distribution gradient $\overrightarrow{g}_\text{dist}$
>
> \begin{aligned}
> \min\_{(x,y)} l\_{\text {adv }}(x, y) =\left|\frac{\nabla l(\theta,(x, y)) \cdot \vec{g}\_{\text {dist }}}{|\nabla l(\theta,(x, y))|\left|\overrightarrow{g}\_\text{dist }\right|}\right| \text{ where }
> \vec{g}\_{\text {dist }}=\frac{1}{|D|} \sum\_{\left(x\_i, y\_i\right) \in D} \nabla l\left(\theta,\left(x\_i, y\_i\right)\right)
> \end{aligned}
>
> Unlike CANIFE, the threat model here indeed is the release of input-space canaries. Therefore, in response to your question, we have implemented Alg 3 and share results on MNIST. We optimize the objective down to $\ell_\text{adv} \leq 0.0001$. We then evaluate the resulting canaries (see updated manuscript)
> 	- Initializing the in-distribution sample (following Line 4 of the algorithm), we achieve 0.2% TPR@0.1FPR.
> 	- Initializing from a canary sampled uniformly at random, and optimizing using Alg. 3, we achieve 7.4% FPR@0.1FPR.
>
> These results are close to in-distribution and adversarial example baselines for MNIST in Table 2, respectively. What these baselines share is the fact that the canary gradients are not being shaped by the model training dynamics. The **average in-distribution gradient in-distribution gradient $\overrightarrow{g}_\text{dist}$ is essentially a constant.**
> Comparing this to our bi-level objective formulation in Eq. 5 (reproduced here for convenience):
>
> \begin{aligned}
> \max\_{(x,y)}\; \ell\_{\text{priv}}(x,y)
> &= f(\theta\_{D\cup\{(x,y)\}};x,y) - f(\theta\_{D};x,y) \\
> \text{s.t.}\quad
> \theta\_{D\cup\{(x,y)\}}
> &\in \arg\min\_\theta \tfrac{1}{|D|+1}  \sum\_{z\_i\in D\cup\{(x,y)\}}  \mathcal{L}(\theta; z\_i), \quad \theta\_{D}
> &\in \arg\min\_\theta \tfrac{1}{|D|}  \sum\_{z\_i\in D}  \mathcal{L}(\theta; z\_i),\nonumber
> \end{aligned}
>
>   Note that the first constraint depends on the canary that is being put in the training set; therefore, a change in the canary $(x, y)$ leads to changes in the resulting model $\theta_{D\cup\{(x,y)\}}$ that needs to be taken into consideration in the unrolled loss. Nasr's $\ell_\text{adv}$ does not take the impact of the inclusion of the canary in the training procedure directly as we do, but rather approximates it by optimizing the proxy objective $\ell_\text{adv}$.
>
> Nasr et al. 2023 justify their choice of the proxy objective empirically, noting in Section B.2," using the dot product between the privatized gradient and the canary gradient is a sufficient metric for auditing DP-SGD." For the sake of the presentation, let us call this the "orthogonality condition."
>
> To the best of our knowledge, Maddock et al. 2023) first derived the orthogonality condition. Notably, in Appendix A, authors connect their objective to the likelihood ratio test and derive the condition under the assumption that "model update follows Gaussian distribution $\mathcal{N}(\mu, \Sigma)$. The sum of $k$ model updates then either follows $\mathcal{N}(k \mu, k \Sigma)$ (without the canary) or $\mathcal{N}(k \mu+\nabla \ell(z), k \Sigma)$ (with the canary), recalling that $u_c \propto \nabla \ell(z)$ ." Under this assumption, Maddock et al. derive the likelihood ratio test between the null $p_0$ and alternative $p_1$ distributions:
>
> > In particular, for the centers of the Gaussian with and without the canary, $u \in\{k \mu, k \mu+\nabla \ell(z)\}$,
> > $$
> \log \left(\frac{p_1(u)}{p_0(u)}\right)= \pm \frac{1}{2} \nabla \ell(z)^T(k \Sigma)^{-1} \nabla \ell(z) .
> $$
> > Maximizing this term will thus help separate the two Gaussians. **However, doing this directly is infeasible as it requires forming and inverting the full covariance matrix $\Sigma$ in very high dimensions.** Instead, we propose to minimize $z \mapsto\left(\nabla \ell(z)^T\right) \Sigma(\nabla \ell(z))$ as it is tractable and can be done with SGD. Note that for sample model updates $\{u_i\}$ we can estimate the (uncentered) covariance matrix as $\frac{1}{n} \sum_i u_i u_i^T$ and thus
> > $$
> \left(\nabla \ell(z)^T\right) \Sigma(\nabla \ell(z)) \approx \frac{1}{n} \sum \nabla \ell(z)^T\left(u_i u_i^T\right) \nabla \ell(z)=\frac{1}{n} \sum\left\langle u_i, \nabla \ell(z)\right\rangle^2 .
> $$
>
> We see that **the key approximation that leads to the orthogonality condition is the inverse-product-Hessian (IVHP)  $\nabla \ell(z)^T(k \Sigma)^{ -1} \nabla \ell(z)$  which Maddock et. al consider infeasible to do with SGD. But this IVHP calculation is exactly what we do efficiently** using influence function in Appendix A (for the IF-OPT baseline); see Eq. 10 in Appendix A of the manuscript; and improve upon using unrolled gradients in the main matter for OptiFluence!

---

### Official Review · Reviewer_5zJV · 2025-10-31

**Soundness:** 3
**Presentation:** 2
**Contribution:** 2
**Rating:** 4
**Confidence:** 4

**Summary:**

This paper claims that existing privacy canaries, like mislabeled or out-of-distribution (OOD) points, are ad hoc and ineffective. It proposes to replace these guesses with a "principled" bilevel optimization framework. The method, OptiFluence, first finds a promising "seed" point from the real data using influence functions (IF-Init) and then uses computationally expensive unrolled optimization (ReMat+TBPTT) to fine-tune the sample's pixels for maximum detectability.

**Strengths:**

1. The core idea is sound. Moving from "guessing" a canary to "optimizing" one is a logical step.
2. The results are undeniable. Table 1 shows that optimized canaries are 415x more detectable than standard ones on CIFAR-10.
3. The transferability result is the paper's strongest practical contribution.

**Weaknesses:**

1. Calling this "scalable" in the title is a serious overstatement. The method is built on unrolled optimization, which is notoriously memory- and compute-intensive. The experiments are confined to MNIST and CIFAR-10. This will not scale to any model we actually care about auditing (e.g., LLMs).
2. Only report TPR @ Low FPR; other metrics such as AUC should also be considered.
3. The baseline of MIA is kind of outdated. There are more recent and powerful MIA attacks such as "Zarifzadeh, Sajjad, Philippe Liu, and Reza Shokri. "Low-cost high-power membership inference attacks." Proceedings of the 41st International Conference on Machine Learning. 2024."
4. The entire optimization objective is to maximize the LiRA "hinge" score. The canaries are overfit to this specific MIA. More MIA attacks should be considered.

**Questions:**

1. The "scalable" claim is tested on CIFAR-10. What is the actual wall-clock time and VRAM cost?  How can you claim this is feasible for large-scale models?
2. For Table 2, what is the performance of DP-SGD?
3. Does your proposed method still work for other MIA?

---

> ### Author Response · Authors · 2025-11-24
>
> We thank the reviewer for their detailed feedback. Since the points raised in the weaknesses and the questions can have a significant overlap, we will be answering them jointly. We use numbered W(eakness) and Q(questions) to refer to the comments. We kindly invite the reviewer to consult the **updated manuscript** while considering our responses, where *additions are marked in blue*, and *fixes are marked in red*.
>
> > W1. Calling this "scalable" in the title is a serious overstatement. The method is built on unrolled optimization, which is notoriously memory- and compute-intensive.
>
> Our claim of scalability is in the context of the relevant baselines. The first being truly-unrolled (exact) canary gradient, which we dubbed Unrolled-OPT (Section 5.2). These are indeed memory-intensive, as we point out in L304:  "When differentiating through many training steps, the computational graph grows linearly with the number of updates, leading to prohibitive memory usage." One of the novelties of OptiFluence, in comparison, is making unrolled gradients computationally feasible through rematerialization (trade-off memory with time) and truncation (trade-off exactness with memory); hence, we have provided knobs for scaling the computational load of optimizing canaries.
>
> > The experiments are confined to MNIST and CIFAR-10. This will not scale to any model we actually care about auditing (e.g., LLMs).
>
> Regarding datasets, we note that CIFAR10 is a standard dataset in privacy auditing of vision models, given the sheer computation necessary to find rare privacy-infringing events [1,2]. Regarding the scalability comments: we have successfully shown that canaries trained for smaller models (ResNet9) are powerful in auditing much larger models (ResNet50, WideResNet).
>
> Finally, auditing language model privacy comes with its own unique challenges, such as a) optimization space being the token-space (embeddings), b) the fact that the notion of the (differential) privacy unit is under-defined for language models("what is a 'sample' in a sequence-to-sequence language model?"). We nevertheless agree that extending our optimization framework to language model auditing is an exciting venue for future work.
>
> Regarding scalability, please also see our response to Reviewer VkF8's W8 and Q9.
>
> [1] Nasr, Milad, Jamie Hayes, Thomas Steinke, Borja Balle, Florian Tramèr, Matthew Jagielski, Nicholas Carlini, and Andreas Terzis. 2023. “Tight Auditing of Differentially Private Machine Learning.” *Proceedings of the 32nd USENIX Conference on Security Symposium* (Anaheim, CA, USA), Sec ’23, 2023.
>
> [2] Nasr, Milad, Shuang Song, Abhradeep Thakurta, Nicolas Papernot, and Nicholas Carlini. 2021. “Adversary Instantiation: Lower Bounds for Differentially Private Machine Learning.” *arXiv:2101.04535 [Cs]*, January 11, 2021. [http://arxiv.org/abs/2101.04535](http://arxiv.org/abs/2101.04535).
>
> > W2. Only report TPR @ Low FPR; other metrics, such as AUC, should also be considered.
>
> We note that we already share ROC curves in Figure 7 in Appendix C.3. We are happy to add AUC numbers (area under these curves) to each figure, if doing so alleviates the reviewer's concern.
>
> The reason we do not already report AUC numbers is that prior work [1] shows that AUC "is often uncorrelated with low false-positive success rates." AUC is considered an average-case success attack metric for the attacker, while TPR@lowFPR is a more stringent worst-case metric. Finally, worst-case TPR, FPR numbers have a direct relationship with privacy parameter lower bounds $\varepsilon_{-}$ (see our response to Reviewer VkF8's W4).  Therefore, we chose to report TPR@lowFPR.
>
> [1] Carlini, Nicholas, Steve Chien, Milad Nasr, Shuang Song, Andreas Terzis, and Florian Tramèr. 2022. “Membership Inference Attacks From First Principles.” May 1, 2022, 1897–1914. [https://doi.org/10.1109/SP46214.2022.9833649](https://doi.org/10.1109/SP46214.2022.9833649).

---

> ### Author Response · Authors · 2025-11-24
>
> > The baseline of MIA is kind of outdated.  There are more recent and powerful MIA attacks, such as "Zarifzadeh, Sajjad, Philippe Liu, and Reza Shokri. "Low-cost high-power membership inference attacks." Proceedings of the 41st International Conference on Machine Learning. 2024."
>
> We respectfully disagree with the statement that our attacks are outdated. We use the LIRA-based attacks introduced in Aerni 2024 et al. (CCS 2024) [1].
>
> We note that **our contribution is not the evaluation (privacy attack)** which we adopted from Aerni 2024  verbatim, but rather the canary to be used for its evaluation. **We successfully show that under a fixed evaluation scheme (i.e., privacy attack), our canaries improve 3 orders of magnitude compared to baselines. A stronger privacy attack (such as the RMIA method from the cited paper) can only improve our already high TPR@lowFPR numbers.**
>
> [1] Aerni, Michael, Jie Zhang, and Florian Tramèr. 2024. “Evaluations of Machine Learning Privacy Defenses Are Misleading.” arXiv:2404.17399. Preprint, arXiv, April 26. [http://arxiv.org/abs/2404.17399](http://arxiv.org/abs/2404.17399).
>
> > The entire optimization objective is to maximize the LiRA "hinge" score. The canaries are overfit to this specific MIA. More MIA attacks should be considered.
>
> The LiRA hinge loss follows from likelihood tests with a prior assumption of Gaussianity (an assumption that, given a large sample size, the central limit theorem well supports).  The Neyman-Pearson lemma establishes that thresholding this statistic is the optimal test. Given the principled and optimal derivation of the prior work, we fail to see the need for using other test statistics that are more heuristic and much less adopted.
>
> Furthermore, attacks give a lower bound on privacy leakage. So it does not really matter if one "overfits" to one attack or not, given that, by definition, we want the strongest possible attack to achieve the best possible lower bound. Since LiRA's statistic is a much stronger attack than other scores (such as confidence values or cross-entropy losses), it makes sense to focus on it.
>
> In response to your and Reviewer VkF8's feedback, in the updated manuscript, we have added a paragraph to further expand the derivation of our privacy loss. See our response to VkF8's Q2.
>
> > Q1: The "scalable" claim is tested on CIFAR-10. What is the actual wall-clock time and VRAM cost?
>
> We added Table 2 in Section 6.2 for a measured wall-cock time and peak VRAM usage in response to the reviewer for both the canary initialization step and the optimization step. We also added a paragraph for the explanation of the scalability of our method.
>
> > Q2: For Table 2, what is the performance of DP-SGD?
>
> We train the non-private CIFAR10 models to 92% accuracy and the private ones to 40-45% accuracy (depending on epsilon). We should clarify that we train these models with relatively few epochs (20), which degrades DP-SGD generalization. We note that tight auditing of DP-SGD is not a focus of our work. We seek to validate the relative performance of our method for different levels of privacy parameter in Table 2—a goal that we achieve. Given the sheer amount of experiments and ablations necessary to validate Optifluence otherwise, we cannot afford to train individual models for 200+epochs, which is necessary to achieve SOTA generalization for DP-SGD on CIFAR10.
>
> We thank the reviewer for their insightful commentary and hope to have answered their questions. If so, we would appreciate it if they increased their score. We welcome any follow-up questions.

---

### Official Review · Reviewer_mvjG · 2025-10-31

**Soundness:** 4
**Presentation:** 4
**Contribution:** 3
**Rating:** 8
**Confidence:** 4

**Summary:**

The paper introduces OptiFluence, which the authors claim is a state of the art privacy canary generation method via a bilevel optimization program that maximizes the likelihood ratio for membership inference by first initializing the canary based on its influence on other samples and then performing an efficient (in compute and memory) gradient-based optimization to fine-tune these samples to maximize the likelihood ratio. They also introduce methods to make their canary optimization efficient in memory and compute by unrolling model updates and checkpointing the computational graph, or even truncating the gradient propagation.

**Strengths:**

**[S1]** Very well-motivated and concretely described methodology, with an attention to detail to practical concerns like compute and memory costs, yielding a practical design that vastly outperforms baselines, which is outstanding.

**[S2]** Use of influence functions to initialize canary is very well motivated and grounded in existing research, and its utility firmly corroborated by ablation studies.

**[S3]** Speaking of which, all the components of the OptiFluence method are covered and ablated in the ablation study section, which very clearly shows each component’s significance. Put another way, I think this is a very well executed ablation study section. In addition, figure 3 provides a good overview of how all the components come together to yield strong canaries (high TPR at very low FPR) as compared to other (ablated) variants.

**[S4]** Good takeaways, viz. pointing out the limitations of mislabeling for canary generation, illustrating how initializing with influence functions provides a more effective and principled approach.

**[S5]** Transferability of generated canaries is a huge positive and contributes to efficient auditing practices.

**[S6]** Auditing (with and without DP-SGD) is well done, with strong choices of auditing methods and well-executed DP-SGD training with a Renyi DP based accountant.

**[S7]** The authors provide an anonymized link to the code and relevant hyperparameters in the appendix, aiding in reproducibility of their results.

**Weaknesses:**

**[W1]** Not a serious weakness/dealbreaker, but it would be desirable to see results on more involved datasets than CIFAR-10 and MNIST. These datasets are popular classic datasets, so to speak, but it would be interesting to see if these results generalize to much larger datasets or datasets with many more classes than 10 (viz. CIFAR-100), more interesting sample distributions, or (this next part is not needed, so the authors can safely ignore this, but it would be appealing) other modalities than image datasets.

**[W2]** Seeing as how attack success (unsurprisingly) degrades for low values of $\varepsilon$ for DP-SGD auditing, could the authors please add experiments on lower values of $\varepsilon$ (viz. 1 and <1)? While these values may yield lower utility of the model, theoretically they are desirable (especially <1) and it would be useful to see how Optifluence (and its baselines) perform in this regime.

**Questions:**

**[Q1]** Can you address W1 and add results on more datasets in different regimes (more classes, samples, different distributions)?

**[Q2]** Can you investigate the efficacy of Optifluence and its baselines in low $\varepsilon$ (high privacy) regimes for DP-SGD auditing?

---

> ### Author Response · Authors · 2025-11-24
>
> We thank the reviewer for their detailed feedback. Since the points raised in the weaknesses and the questions can have a significant overlap, we will be answering them jointly. We use numbered W(eakness) and Q(questions) to refer to the comments. We kindly invite the reviewer to consult the **updated manuscript** while considering out responses where *additions are marked in blue*, and *fixes are marked in red*.
>
> > W1/Q1: ... It would be interesting to see if these results generalize to much larger datasets or datasets with many more classes than 10 (viz. CIFAR-100)
>
> In Table1, we have updated the results for CIFAR100 using our method as well, which achieved also nearly perfect detectability, showing our method is able to generalize to more challenging and complex datasets.
>
> > W2/Q2: ...Could the authors please add experiments on lower values of $\varepsilon$ (viz. 1 and <1)? While these values may yield lower utility of the model, theoretically they are desirable (especially <1) and it would be useful to see how Optifluence (and its baselines) perform in this regime.
>
> We ran additional experiments for $\varepsilon \in \{0.5, 1\}$ with CIFAR10 and updated Table 3 accordingly. We get 1.6% TPR@0.1FPR for $\varepsilon=1$ but does the attack is not successful for $\varepsilon=0.5$ . We should emphasize that these $\varepsilon$ values are very low for vision models, and CIFAR10 models in particular. In the literature, it is common to audit CIFAR10 models trained with $\varepsilon=8$ . Therefore we do not find these results surprising.
>
> As we discuss in Response W2 to Reviewer VkF8's Q5, tight DP-Auditing often requires *canary gradients* or manipulation of the training procedure. Input-space canaries therefore are at an inherent disadvantage. However, they enable third-party auditing, are architecture-agnostic, and transfer. We nevertheless have included DP-Auditing results in Section 6.2 (Table 3) to show **where first-party tight audits are not possible optimized, canaries can provide a useful alternative.**
>
> > Q3: The other reviewers rightly point out the need to justify and corroborate the efficiency of unrolled updates in your paradigm with empirical evidence (runtime and memory used) and on more involved/expensive settings than CIFAR-10 and MNIST. Can the authors please address that?
>
> To answer the reviewer’s comment regarding empirical evidences for efficiency of out method, we have updated Section 6.2 to report the requested metrics and added a paragraph explaining why our method scales effectively from a small dataset such as MNIST to a more complex dataset like CIFAR-100. We have also updated Table 1, which now shows near-perfect detectability on CIFAR-100 as well.
>
> We thank the reviewer for their insightful commentary and hope to have answered their questions. If so, we would appreciate it if they increased their score. We welcome any follow-up questions.

---

> > ### Comment · Reviewer_mvjG · 2025-11-24
> >
> > Many thanks to the authors for their response!
> >
> > I think that these added results further strengthen the paper and provide a broader evaluation over several privacy regimes  and a more involved dataset and concretely illustrates the efficiency of their method from a runtime and memory perspective.
> >
> > The paper thus retains my strong support, as stated above. My score already reflects strong support for the paper; I will have to think about whether I'll increase my score to a perfect, award-worthy 10 will be based on the other reviewer's feedback and addressing of remaining concerns (and the panel's consensus), etc.
> >
> > Regardless, I will firmly champion the paper's acceptance, and wish the authors all the very best and congratulate them on a valuable submission to ICLR and for the privacy community.

---

### Official Review · Reviewer_VkF8 · 2025-11-01

**Soundness:** 2
**Presentation:** 3
**Contribution:** 2
**Rating:** 4
**Confidence:** 3

**Summary:**

The paper proposes a framework for constructing privacy canaries—artificial samples used to audit data leakage in machine learning models. Unlike prior methods that rely on mislabeled or out-of-distribution samples, OptiFluence formulates canary design as a bilevel optimization problem that maximizes the likelihood-ratio statistic used in membership inference attacks, effectively identifying samples that most alter the model’s behavior when included in training. The framework combines two key components: IF-Init, which uses influence functions to pre-select rare, highly self-influential samples likely to be memorized, and Unrolled-Opt, which refines these candidates by differentiating through the training process using memory-efficient techniques such as rematerialization and truncated backpropagation through time (TBPTT). Experiments on MNIST and CIFAR-10 demonstrate that OptiFluence achieves near-perfect detectability (up to 99.8% TPR at 0.1% FPR), outperforming baselines by up to 415×.

**Strengths:**

1. The proposed OptiFluence pipeline combines influence-based initialization (IF-Init) with unrolled optimization enhanced by rematerialization and truncated backpropagation (ReMat+TBPTT) for scalability.

2. Achieves near-perfect canary detectability (up to 99.8% TPR at 0.1% FPR) and up to 415× improvement over heuristic baselines on MNIST and CIFAR-10.

3. Optimized canaries generalize well across architectures (e.g., ResNet-9 → ResNet-50), enabling efficient third-party or regulatory auditing without model retraining.

4. Includes detailed ablation studies, comparisons to prior methods (e.g., metagradient-based optimization), and evaluations under DP-SGD training.

5. The paper is well-structured, with a clear presentation of motivation, methodology, and results.

**Weaknesses:**

1. The files at the anonymous link do not open. Please either include a zip file with the source code and scripts in the supplementary material or update the link to ensure the files are accessible.

2. The central idea—to optimize the log-likelihood ratio via a bilevel objective—is elegant but remains largely heuristic. The paper argues that maximizing the difference in logits between models trained with and without the canary approximates the true likelihood-ratio test (Equation 5), yet provides no formal proof that this surrogate objective is consistent or unbiased. Without theoretical guarantees or approximation bounds, it is unclear whether the optimized canary truly maximizes detectability in general or only within the specific experimental setup.

3.  The bilevel objective requires differentiating through the entire training trajectory, but the paper does not discuss convergence, stability, or variance of this optimization—particularly when truncated backpropagation and rematerialization are used. The claim in Section 6.3 that unrolled optimization yields “exact gradients” appears overstated, since ReMat+TBPTT introduces gradient truncation and therefore only provides approximate updates. It would strengthen the paper to quantify how this approximation impacts final canary detectability.

4.  While the paper references differential privacy definitions and ε-bounds (Equation 2), it never formally connects the empirical detectability metric (TPR@FPR) to theoretical privacy parameters ε or δ. The framework is therefore empirical rather than analytical, and the paper should make this distinction explicit.

5.  Because OptiFluence directly maximizes the LiRA hinge-based likelihood-ratio score, the optimized canaries may overfit to this particular attack formulation. The paper does not evaluate the canaries under alternative membership inference metrics (e.g., confidence-, entropy-, or loss-based scores), leaving open the question of whether detectability generalizes to unseen auditing methods.

6. The reported transferability of canaries across architectures (e.g., ResNet-9 → ResNet-50) is intriguing but lacks theoretical explanation. The paper attributes it qualitatively to shared representation geometry, yet provides no analysis of why optimized samples remain highly distinguishable under different model dynamics. This weakens the claim that OptiFluence supports “third-party audits without retraining.”

7. Figures 4 and 6 show that optimized canaries—particularly on CIFAR-10—often appear visually unnatural or out-of-distribution. This raises the possibility that their detectability stems from atypical low-level statistics rather than genuine memorization.

8. The method involves repeated model retraining with unrolled optimization, rematerialization, and truncated backpropagation—all computationally demanding procedures. However, the paper does not report runtime, GPU memory usage, or total training cost. Without this information, it is difficult to assess whether the framework is truly scalable beyond small benchmark datasets.

Minor comments:

1. The acronym ERM (Empirical Risk Minimization) is used without definition and should be introduced upon first mention.

**Questions:**

1.  The anonymous code link provided in the submission does not open. Could you please share a working repository or include a zip file in the supplementary material to ensure full reproducibility?

2. Could you provide a more rigorous justification for treating the logit difference (Equation 5) as a valid surrogate for the likelihood-ratio statistic? Specifically, under what assumptions does maximizing this surrogate guarantee improved membership distinguishability, and can any theoretical bound or consistency argument be established?

3. The paper claims that unrolled optimization provides “exact gradients,” yet the use of truncated backpropagation and rematerialization implies an approximation. Could you quantify how this truncation affects the final canary detectability? For instance, how does TPR@FPR vary as the truncation window K changes?

4. What measures were taken to ensure optimization stability across seeds and models? Do different initialization points (e.g., influence-selected seeds vs. random) lead to consistent canary detectability, or is the outcome highly variable?

5. Since the framework is inspired by differential privacy but ultimately empirical, can you clarify how the metric (TPR@FPR) relates to formal ε or δ values? Is there any attempt to estimate lower bounds on ε or compare to DP auditing baselines that produce numeric privacy budgets?

6. The optimization directly targets the LiRA hinge-based statistic. Would the optimized canaries remain highly detectable under alternative membership inference attacks (e.g., confidence-, entropy-, or loss-based)? Have you evaluated cross-auditor robustness?

7. The transferability of optimized canaries across architectures is one of the key selling points of the paper. Can you provide a theoretical or empirical explanation for why canaries optimized on ResNet-9 remain highly detectable on ResNet-50 or WideResNet? Is this phenomenon architecture-dependent or data-dependent?

8.  Some optimized canaries, particularly in CIFAR-10, appear visually unnatural or off-manifold. Have you attempted to quantify the degree of deviation from the data distribution (e.g., via FID, nearest-neighbor distance, or classifier confidence)? Could detectability be driven by such distributional shifts rather than true memorization?
9.  The paper emphasizes scalability, yet unrolled optimization and rematerialization are computationally heavy. Could you please report the runtime, GPU memory footprint, and training cost for the CIFAR-10 experiments? How would the approach scale to larger datasets or transformer architectures?

---

> ### Author Response · Authors · 2025-11-24
>
> We thank the reviewer for their detailed feedback. Since the points raised in the weaknesses and the questions can have a significant overlap, we will be answering them jointly. We use numbered W(eakness) and Q(questions) to refer to the comments. We kindly invite the reviewer to consult the **updated manuscript** while considering our responses, where additions are marked in blue, and fixes are marked in red.
>
> > W1/Q1: The files at the anonymous link do not open.
>
> We apologize for the link not working. The Open Science service we used has gone offline, making the code unavailable. In the rebuttal revision, we have submitted a zipped file for the code.
>
> > W2: ...Without theoretical guarantees or approximation bounds, it is unclear whether the optimized canary truly maximizes detectability in general or only within the specific experimental setup.
>
> We agree with the reviewer that formal results would be interesting. However, the lack of general bounds is expected given the bi-level characterization of the problem with the training loss objective (of a multi-million parameter neural network) in the constraint set. Even the state-of-the-art optimization results for neural networks are limited to a few-layer networks.
>
> On the topic of "heuristic" approaches: previously, canaries had to be *hand-crafted* as there was no systematic way to generate canaries. **The important novelty of our work is that we can _automatically_ generate canaries that outperform hand-crafted ones.**
>
> Our work empirically demonstrates the practical feasibility of optimized canaries as a framework. We do agree that formal results would be intriguing, but given the lack of relevant literature and the density of the formalization and modeling already present in the paper, we intend to pursue formal results in future work.
>
> > Q2: Could you provide a more rigorous justification for treating the logit difference (Equation 5) as a valid surrogate for the likelihood-ratio statistic? Specifically, under what assumptions does maximizing this surrogate guarantee improved membership distinguishability
>
> Of course:
> > [**From Updated Manuscript**]
> Carlini et al. [1, Section IV.C] provide a justification for using logit-scaled confidence values instead of their unscaled values, or even the cross-entropy loss. Notably, in Figure 4, it is shown that **logit-scaled confidence values provide a more Gaussian distribution. Gaussianity is important because it allows a more efficient parametric modeling of null and alternative distributions** $Q_\text{in}$ and $Q_\text{out}$. In our work, we optimize these parametric distributions by optimizing the canary sample, and preserving Gaussianity is equally important to us.
> Finally, in Section VI.A, it is shown that in a neural network that produces pre-softmax (i.e. unnormalized) values $g(x)$, logit-scaled confidence $\phi(\frac{p}{1-p} )$ where $\phi$ is the logit function, and $p$ are confidence scores such that $p = \operatorname{softmax}(g(x))$. These logit-scaled confidence values can be calculated with more numerical stability using the "hinge" loss $g(\theta;x)\_y - \operatorname{LogSumExp}\_{y'} g(\theta;x)\_{y'}$.
>
> We initially did not include the above discussion as the original work tackles the question of test statistic modeling and efficient evaluation in much broader detail. But we acknowledge that having this description would clarify the derivation, and have since added it to the paper.
>
> > W.3 ...The claim in Section 6.3 that unrolled optimization yields “exact gradients” appears overstated, since ReMat+TBPTT introduces gradient truncation and therefore only provides approximate updates.
>
> We do not use the term "exact gradient" lightly. **Our "unrolled" baseline on MNIST does indeed take a hyper-gradient that is exact in the sense that we create the complete training run as a single computational graph** and differentiate through it using automatic differentiation. This is not scalable to larger datasets that need larger models, which is why ReMat and TBPTT were introduced. **We like to emphasize that the use of ReMat alone also does not trade off with the accuracy of the gradients, as we only trade off memory for time.** ReMat is now a standard feature in our software suite `JAX` and is handled by its compiler `XLA`, which shows exactness is not in question.
>
> To respond to the reviewer's point about the impact of truncation, we point the reviewer to our new CIFAR100 experiment, which required us to reduce $k=4$ from CIFAR10 experiments to $k=2$ in order to fit in memory. The results are reflected in the **updated Table 1,** where we show that our CIFAR100 canaries (despite having been trained with lower number of training steps) achieved near perfect detectability.

---

> ### Author Response · Authors · 2025-11-24
>
> > Q3: ...Could you quantify how this truncation affects the final canary detectability? For instance, how does TPR@FPR vary as the truncation window K changes?
>
> In practice, we did not observe a meaningful impact to tuning $k$ . We have added Figure 12 in Appendix C.7 to showcase the effect of $k$ on the canary loss where we compare "no truncation," $k=4$ and $k=2$ for MNIST. However, **lower $k$ does not in fact impact the TPR@lowFPR scores of our attacks. As even canaries trained with $k=2$ remain nearly perfectly detectable.**
>
> Therefore, **we treat $k$ as not ML hyper-parameter that needs to be tuned.** Our suggestion to the practitioners is to use the largest $k$ that allows them to train accurate models with the memory they have access to; since larger $k$ by definition indicates a more accurate canary gradient; but that additional accuracy may not necessary to achieve highly detectable canaries.
>
> > W4: While the paper references differential privacy definitions and ε-bounds (Equation 2), it never formally connects the empirical detectability metric (TPR@FPR) to theoretical privacy parameters ε or δ. The framework is therefore empirical rather than analytical
>
> The theoretical privacy parameters $(\varepsilon)$ is an upperbound to the privacy loss that we models, approximate and optimize as we show in Eq. (2) and disucss extensively in Lines 153–164. Furthermore, we do report these upperbounds in our DP-Auditing result in Table 2. Therefore, we interpret the reviewer's comments to be about  $\varepsilon$ *lower bounds* which are empirical quantities.
>
> There is indeed a long list of works that have already tackled the calculation of these lower bounds from the results of membership inference attacks (MIAs). For example, from Zanella-Béguelin et al.[1]:
> $$\hat{\varepsilon}_{-}=\max \\{\log \frac{1-\delta-\mathrm{FPR}}{\mathrm{FNR}}, \log \frac{1-\delta-\mathrm{FNR}}{\mathrm{FPR}}\\}$$
> where FPR and FNR (1-TPR) of MIAs are estimated using a Monte Carlo approach.
>
> These empirical quantities can be calculated for MIAs against any training algorithm—even non-DP ones that have no theoreitical privacy accounting for $\varepsilon$ . Therefore, there is no inherent benefit in going through the proxy of $\varepsilon$ lowerbounds to present MIA attack socres. In fact, new work [2] shows that privacy risks and mitigations can be formalized directly within a privacy-attack framing.
>
> [1] Zanella-Béguelin, Santiago, Lukas Wutschitz, Shruti Tople, Ahmed Salem, Victor Rühle, Andrew Paverd, Mohammad Naseri, and Boris Köpf. 2022. Bayesian Estimation of Differential Privacy. arXiv:2206.05199. arXiv. https://doi.org/10.48550/arXiv.2206.05199.
>
> [2] Kulynych, Bogdan, Juan Felipe Gomez, Georgios Kaissis, Flavio du Pin Calmon, and Carmela Troncoso. 2024. “Attack-Aware Noise Calibration for Differential Privacy.” arXiv:2407.02191. Preprint, arXiv, November 7. https://doi.org/10.48550/arXiv.2407.02191.

---

> ### Author Response · Authors · 2025-11-24
>
> > Q5. Is there any attempt to estimate lower bounds on ε or compare to DP auditing baselines that produce numeric privacy budgets?
>
> We have addressed this question in W4. We want to additionally note that "DP Auditing" baselines often go beyond auditing with canary samples and produce canary gradients [1] or otherwise change the training procedure [2] to achieve the tightest lower bounds possible. **Our work is strictly in the space of auditing using canaries in the input (sample) space. The benefit of this type of audit is its versatility (given its architecture-agnosticism) and transferability (which we have demonstrated).**
>
> We nevertheless have included DP-Auditing results in Section 6.2 (Table 3) with the TPR@0.1FPR metric to show that **where first-party tight audits are not possible, optimized canaries can provide a useful alternative.** Additionally, to answer the reviewer's question, using the [`privacy-estimates`](https://github.com/microsoft/responsible-ai-toolbox-privacy) by Zanella-Béguelin et al, we have calculated the following lower bounds for an MNIST model trained with DP-SGD with $\varepsilon \in \\{1, 2, 6, 8\\}$ and measured the lower bounds.
>
>  \
> *DP-SGD Audits for models trained on MNIST, with measured  TPR@0.1FPR and  lower bounds  $\hat\varepsilon_{-}$*
> | $\varepsilon$ | TPR@0.1FPR | $\hat\varepsilon_{-}$ |
> |---|---|---|
> | 0.5 | 0.10 | $(0, \infty)$ |
> | 1 | 0.13 | $(0, \infty)$ |
> | 2 | 0.26 | $(0, \infty)$ |
> | 6 | 0.31 | $(0.26, 0.93)$ |
> | 8 | 0.32 | $(0.35, 0.74)$ |
>
>
>
> We note that due to the nature of these algorithms, the lower bounds estimation itself comes as a confidence interval that shrinks with a higher number of shadow models.  We see that for small theoretical epsilons ($\varepsilon \leq 2$), we cannot estimate usable lower bounds, but TPR@0.1FPR values remain useful. It is possible that with a higher number of shadow models (here we used 20k as in the paper), we may be able to estimate non-vacuous lower bounds. But even this example shows that estimating lower bounds adds yet another layer of complexity to characterizing the privacy leakage of the models.
>
> [1] Nasr, Milad, Jamie Hayes, Thomas Steinke, Borja Balle, Florian Tramèr, Matthew Jagielski, Nicholas Carlini, and Andreas Terzis. 2023. “Tight Auditing of Differentially Private Machine Learning.” *Proceedings of the 32nd USENIX Conference on Security Symposium* (Anaheim, CA, USA), Sec ’23, 2023.
>
> [2] Nasr, Milad, Shuang Song, Abhradeep Thakurta, Nicolas Papernot, and Nicholas Carlini. 2021. “Adversary Instantiation: Lower Bounds for Differentially Private Machine Learning.” *arXiv:2101.04535 [Cs]*, January 11, 2021. [http://arxiv.org/abs/2101.04535](http://arxiv.org/abs/2101.04535).
>
> > W5/Q6: ...OptiFluence directly maximizes the LiRA hinge-based likelihood-ratio score; the optimized canaries may overfit to this particular attack formulation. The paper does not evaluate the canaries under alternative membership inference metrics (e.g., confidence-, entropy-, or loss-based scores)...
>
> The LiRA hinge loss follows from likelihood tests with a prior assumption of Gaussianity (an assumption that, given a large sample size, the central limit theorem well supports).  The Neyman-Pearson lemma establishes that thresholding this statistic is the optimal test. Given the principled and optimal derivation of the prior work, we fail to see the need for using other test statistics that are more heuristic and much less adopted.
>
> Furthermore, attacks give a lower bound on privacy leakage. So it does not really matter if one "overfits" to one attack or not, given that, **by definition, we want the strongest possible attack to achieve the best possible lower bound. Since LiRA is a much stronger attack than the others (Carlini et al 2022 discuss and evaluate this point at length), it makes sense to focus on it.**
>
> > W6: ...The reported transferability of canaries across architectures (e.g., ResNet-9 → ResNet-50) is intriguing but lacks a theoretical explanation. *The paper attributes it qualitatively to shared representation geometry*...
>
> We are confused about this statement. **We have not made a claim about a "shared representation geometry."** Can the reviewer kindly let us know from what part of the paper they have construed this? We are happy to adjust language to avoid any such confusion.

---

> ### Author Response · Authors · 2025-11-24
>
> > W6:... yet provides no analysis of why optimized samples remain highly distinguishable under different model dynamics.
>
> As discussed in our answer to W2, **given the absence of formal results for bi-level optimization problems for large neural networks, we cannot make meaningful theoretical claims. This is even more true for the phenomenon of transferability where the shape of the hypothesis class (weight space) is different between optimization to evaluation.** Therefore, we do not make any theoretical claim about transferability but validate our transferability claim empirically.
>
> We agree that formal characterization of transferability is an exciting venue for future work. However, to the best of our knowledge, **in prior cases of  transferability, such as adversarial examples, the community has not been able to characterize a theoretical reasoning for this phenomenon either. Yet it is useful to know that this property exists and it has been studied empirically extensively as a result.**
>
> > Q7: ... Can you provide a theoretical or empirical explanation for why canaries optimized on ResNet-9 remain highly detectable on ResNet-50 or WideResNet? Is this phenomenon architecture-dependent or data-dependent?
>
> We provide empirical transferability results in Section 6.2, paragraph "Optimized canaries transfer between architectures" which include results for both CIFAR10 (the architectures mentioned) and MNIST (different model widths in Appendix C.4). So our answer is yes.
>
> We can provide the following observational explanation of this phenomenon:
>
> > All models (i.e. hypothesis classes) seek to learn the same concept  from the data. A transferable canary indicates that the notion of a canary is not a function of the of the hypothesis class, but rather the concept class itself. For example, for digit classification, we know that a 2 and a 7 are reasonably close to each other; and written in a bad handwriting, one can be mistaken for the other. Therefore, a good canary can be an image that can reasonably be classified as either 2 or a 7 by even a human—an entirely different learner!
>
> We like to note however that the space of canaries are much larger than the above example. But the above should be sufficient to show why transferability of canaries would make sense in the first place.
>
> > W7: Figures 4 and 6 show that optimized canaries—particularly on CIFAR-10—often appear visually unnatural or out-of-distribution. This raises the possibility that their detectability stems from atypical low-level statistics rather than genuine memorization.
>
> We understand memorization is an umbrella term used in different context in machine learning. However, we respectfully disagree with the reviewer that learning "atypical low-level statistics" is not "genuine memorization." Since ultimately it does not matter how the model overfits to training samples, as long as it does and we can detect it.
>
> **A canary, under the DP definition, does not need to come from a particular data distribution at all.** As long as the model's behavior detectably changes in response to the existence of the canary in the training set, it fits the definition. **We note that prior work in privacy auditing uses far more unnatural-looking canaries.** See [1] for example where a "a 5x5 white square in the top-left corner of an image" is used. Therefore, not even prior work acknowledges the need for naturalness of privacy canaries for auditing purposes.
>
> [1] Jagielski, Matthew, Jonathan Ullman, and Alina Oprea. 2020. “Auditing Differentially Private Machine Learning: How Private Is Private SGD?” *Advances in Neural Information Processing Systems* 33. [https://papers.nips.cc/paper/2020/hash/fc4ddc15f9f4b4b06ef7844d6bb53abf-Abstract.html](https://papers.nips.cc/paper/2020/hash/fc4ddc15f9f4b4b06ef7844d6bb53abf-Abstract.html).
>
> > Q8:...Have you attempted to quantify the degree of deviation from the data distribution (e.g., via FID, nearest-neighbor distance, or classifier confidence)?
>
> We have not. Our only goal is to produce input canaries that fit the DP definition. "Visual naturalness" is therefore not within our desiderata. We publish canary figures to gain an intuitive understanding. One of these understanding is precisely that us humans cannot necessary find a pattern that turns out to be the best canary for privacy auditing.

---

> ### Author Response · Authors · 2025-11-24
>
> > W8: ...The paper does not report runtime, GPU memory usage, or total training cost.
>
> To address the reviewer’s concern about computational efficiency, we have updated Section 6.2 to include **Table 2, which reports both wall-clock time and VRAM usage.** We also added a paragraph explaining why our method scales effectively from a small dataset like MNIST to a more complex one such as CIFAR-100. In particular, to fit CIFAR-100 training in memory, we reduced the TBPTT truncation parameter k from 4 to 2—this parameter controls how many training steps are included in the canary-gradient computation (see Fig. 2(c)). Despite this reduction, the optimized canary still achieved near-perfect detectability on CIFAR-100, further illustrating the scalability of our approach.
>
> > Q9: How would the approach scale to larger datasets or transformer architectures?
>
> Our final design for OptiFluence has several characteristics that simplifies scaling challenges. a) modularity; b) scalability knobs with truncation and influence calculation using EK-FAC approximations (which are scaled to transformers Studying Large Language Model Generalization with Influence Functions); c) first-party privacy auditing has a significant overhead. By showing transferability, one cost is amortized to multiple models; and even multiple parties. For a concrete example of this, see our response to W3.
>
> > Minor comment 1: The acronym ERM (Empirical Risk Minimization) is used without definition and should be introduced upon first mention.
>
> We have updated the manuscript to explain the model is achieved as the result of the minimization of the empirical loss.
>
> We thank the reviewer for their insightful commentary and hope to have answered their questions. If so, we would appreciate it if they increased their score. We welcome any follow-up questions.

---

### Author Response · Authors · 2025-12-03
**Rebuttal Summary**

We thank all reviewers for their thorough and constructive feedback. We are encouraged that **all reviewers recognized the core strengths of OptiFluence**: the principled bilevel optimization framework, the strong empirical results (up to 99.8% TPR@0.1FPR and 415× improvement over baselines), the novel combination of influence-based initialization with unrolled optimization, and particularly the **transferability of optimized canaries across architectures**—which multiple reviewers highlighted as a significant practical contribution enabling efficient third-party auditing.

## Substantial Additions During Rebuttal
  We have significantly strengthened the manuscript through the following additions:

### 1. Extended Experimental Validation

**New CIFAR-100 results** (Table 1) demonstrating near-perfect detectability (99%+) on a more complex dataset with 10× more classes, addressing concerns from Reviewers mvjG and VkF8 about generalization

**Additional low-ε experiments** (ε ∈ {0.5, 1}) for DP-SGD auditing (Table 3), as requested by Reviewer mvjG

**Computational efficiency metrics** in new Table 2, reporting wall-clock time and VRAM usage for both initialization and optimization phases, addressing requests from Reviewers VkF8, mvjG, 5zJV, and 5uPC

**Initialization ablation study** showing IF-Init substantially outperforms random ID-Init (MNIST: 0.998 vs. 0.704; CIFAR-10: 0.995 vs. 0.365), clarifying concerns raised by Reviewer 5uPC

### 2. Technical Clarifications and New Content

**New Appendix D** with extensive technical discussion and empirical implementation of Nasr et al.'s Algorithm 3, definitively positioning our contribution relative to prior work (Reviewers 5uPC and VkF8)

**Enhanced justification** for the logit difference surrogate, incorporating Carlini et al.'s Gaussianity discussion and Neyman-Pearson optimality (Reviewer VkF8, Q2)

**New Figure 12** demonstrating the effect of the truncation parameter k on canary optimization loss. We show show that even k=2 achieves near-perfect detectability (Reviewer VkF8, Q3)

**Detailed scalability discussion** explaining how rematerialization and truncation provide practical knobs for memory-time trade-offs (Reviewers 5zJV and VkF8)

### 3. Clarified Positioning and Methodology

**Input-space vs. weight-space canaries**: We clarified that while prior optimization-based work Maddock et al. also takes place in the input-space; their canaries are released in the weight-space (due to the federated learning setting). OptiFluence however exclusively generates input-space canaries enabling architecture-agnostic, transferable third-party auditing—a fundamentally different threat model and deployment context. We provided an extensive comparison with Maddock et al. 2023 and Nasr et al. 2023 in Appendix D.

**"Exact gradients" terminology**: We explained this refers to our unrolled baseline using complete computational graphs via automatic differentiation. ReMat trades memory for time without sacrificing gradient accuracy, while TBPTT provides a tunable approximation that empirically maintains effectiveness.

**Relationship to adversarial examples**: We emphasized Table 1 already shows adversarial examples perform poorly (0.45% vs. our 99.8%). Our derivation yields adversarial-like gradients with **negative sign**: canaries must remain sensitive during training, unlike adversarial examples designed for robustness.

**Overfitting concerns**: Since our goal is the tightest possible privacy lower bound, maximizing detectability under the strongest principled attack (derived from Neyman-Pearson optimality) is appropriate by definition.

## Remaining Limitations and Future Work

We acknowledge that **formal theoretical characterization** of both the bilevel optimization guarantees and transferability phenomenon remains future work, as noted by Reviewers VkF8 and 5zJV. However, this is expected given the complexity of bilevel optimization over multi-million parameter networks—even state-of-the-art results are limited to small networks. We note that analogous phenomena (e.g., adversarial example transferability) lack formal characterization yet remain extensively studied empirically.

We also acknowledge that **scaling to large language models** presents unique challenges (token-space optimization, undefined privacy units for sequences) that extend beyond the scope of this work focused on vision models, as noted by Reviewer 5zJV.

## Conclusion
  Our rebuttal demonstrates OptiFluence's robustness through additional datasets, lower privacy budgets, and comprehensive ablations. The automatic generation of highly detectable, transferable canaries represents a significant advance over hand-crafted approaches, with practical implications for regulatory and third-party privacy auditing. We believe these substantial additions address all major reviewer concerns and hope reviewers will consider increasing their scores accordingly.

---

### Meta-Review · Area_Chair_3BXp · 2026-01-06

**Summary:**

The paper introduces OptiFluence for privacy canary generation, which is based on a formulation of bilevel program. Although the title claimed "scalable", the scale of the experiments is limited. Additionally,  the theoretical guarantee of proposed method is insufficient. Based on above reasons, I recommend rejection.

**Reviewer Concerns:**

The following main concerns have not been well addressed:
1. It is unclear whether the optimized canary truly maximizes detectability in general.
2. The experiments are confined to MNIST and CIFAR-10.

**Reviewer Scores:**

Since the main concerns have not been addressed, I think the reviewers will keep their scores.

---

### Decision · Program_Chairs · 2026-01-26

Reject